# Greedy Algorithms for Cone Constrained Optimization with Convergence Guarantees

**Francesco Locatello**
MPI for Intelligent Systems - ETH Zurich
locatelf@ethz.ch

**Michael Tschannen**
ETH Zurich
michaelt@nari.ee.ethz.ch

**Gunnar Rätsch**
ETH Zurich
raetsch@inf.ethz.ch

**Martin Jaggi**
EPFL
martin.jaggi@epfl.ch

## Abstract

Greedy optimization methods such as Matching Pursuit (MP) and Frank-Wolfe (FW) algorithms regained popularity in recent years due to their simplicity, effectiveness and theoretical guarantees. MP and FW address optimization over the *linear span* and the *convex hull* of a set of atoms, respectively. In this paper, we consider the intermediate case of optimization over the *convex cone*, parametrized as the conic hull of a generic atom set, leading to the first principled definitions of non-negative MP algorithms for which we give explicit convergence rates and demonstrate excellent empirical performance. In particular, we derive sublinear ($\mathcal{O}(1/t)$) convergence on general smooth and convex objectives, and linear convergence ($\mathcal{O}(e^{-t})$) on strongly convex objectives, in both cases for general sets of atoms. Furthermore, we establish a clear correspondence of our algorithms to known algorithms from the MP and FW literature. Our novel algorithms and analyses target general atom sets and general objective functions, and hence are directly applicable to a large variety of learning settings.

## 1 Introduction

In recent years, greedy optimization algorithms have attracted significant interest in the domains of signal processing and machine learning thanks to their ability to process very large data sets. Arguably two of the most popular representatives are Frank-Wolfe (FW) [12, 21] and Matching Pursuit (MP) algorithms [34], in particular Orthogonal MP (OMP) [9, 49]. While the former targets minimization of a convex function over *bounded convex sets*, the latter apply to minimization over a *linear subspace*. In both cases, the domain is commonly parametrized by a set of atoms or dictionary elements, and in each iteration, both algorithms rely on querying a so-called *linear minimization oracle* (LMO) to find the direction of steepest descent in the set of atoms. The iterate is then updated as a *linear* or *convex combination*, respectively, of previous iterates and the newly obtained atom from the LMO. The particular choice of the atom set allows to encode structure such as sparsity and non-negativity (of the atoms) into the solution. This enables control of the trade-off between the amount of structure in the solution and approximation quality via the number of iterations, which was found useful in a large variety of use cases including structured matrix and tensor factorizations [50, 53, 54, 18].

In this paper, we target an important "intermediate case" between the two domain parameterizations given by the *linear span* and the *convex hull* of an atom set, namely the parameterization of the optimization domain as the *conic hull* of a possibly infinite atom set. In this case, the solution can be represented as a *non-negative* linear combination of the atoms, which is desirable in many

applications, e.g., due to the physics underlying the problem at hand, or for the sake of interpretability. Concrete examples include unmixing problems [11, 16, 3], model selection [33], and matrix and tensor factorizations [4, 24]. However, existing convergence analyses do not apply to the currently used greedy algorithms. In particular, all existing MP variants for the conic hull case [5, 38, 52] are not guaranteed to converge and may get stuck far away from the optimum (this can be observed in the experiments in Section 6). From a theoretical perspective, this intermediate case is of paramount interest in the context of MP and FW algorithms. Indeed, the atom set is not guaranteed to contain an atom aligned with a descent direction for all possible suboptimal iterates, as is the case when the optimization domain is the linear span or the convex hull of the atom set [39, 32]. Hence, while conic constraints have been widely studied in the context of a manifold of different applications, none of the existing greedy algorithms enjoys explicit convergence rates.

We propose and analyze new MP algorithms tailored for the minimization of smooth convex functions over the conic hull of an atom set. Specifically, our key contributions are:

- We propose the first (non-orthogonal) MP algorithm for optimization over conic hulls guaranteed to converge, and prove a corresponding *sublinear* convergence rate with *explicit constants*. Surprisingly, convergence is achieved without increasing computational complexity compared to ordinary MP.

- We propose new away-step, pairwise, and fully corrective MP variants, inspired by variants of FW [28] and generalized MP [32], respectively, that allow for different degrees of weight corrections for previously selected atoms. We derive corresponding sublinear and linear (for strongly convex objectives) convergence rates that solely depend on the geometry of the atom set.

- All our algorithms apply to general smooth convex functions. This is in contrast to all prior work on non-negative MP, which targets quadratic objectives [5, 38, 52]. Furthermore, if the conic hull of the atom set equals its linear span, we recover both algorithms and rates derived in [32] for generalized MP variants.

- We make no assumptions on the atom set which is simply a subset of a Hilbert space, in particular we do not assume the atom set to be finite.

Before presenting our algorithms (Section 3) along with the corresponding convergence guarantees (Section 4), we briefly review generalized MP variants. A detailed discussion of related work can be found in Section 5 followed by illustrative experiments on a least squares problem on synthetic data, and non-negative matrix factorization as well as non-negative garrote logistic regression as applications examples on real data (numerical evaluations of more applications and the dependency between constants in the rate and empirical convergence can be found in the supplementary material).

**Notation.** Given a non-empty subset $\mathcal{A}$ of some Hilbert space, let $\mathrm{conv}(\mathcal{A})$ be the convex hull of $\mathcal{A}$, and let $\mathrm{lin}(\mathcal{A})$ denote its linear span. Given a closed set $\mathcal{A}$, we call its diameter $\mathrm{diam}(\mathcal{A}) = \max_{\mathbf{z}_1, \mathbf{z}_2 \in \mathcal{A}} \|\mathbf{z}_1 - \mathbf{z}_2\|$ and its radius $\mathrm{radius}(\mathcal{A}) = \max_{\mathbf{z} \in \mathcal{A}} \|\mathbf{z}\|$. $\|\mathbf{x}\|_{\mathcal{A}} := \inf\{c > 0 \colon \mathbf{x} \in c \cdot \mathrm{conv}(\mathcal{A})\}$ is the atomic norm of $\mathbf{x}$ over a set $\mathcal{A}$ (also known as the gauge function of $\mathrm{conv}(\mathcal{A})$). We call a subset $\mathcal{A}$ of a Hilbert space symmetric if it is closed under negation.

## 2 Review of Matching Pursuit Variants

Let $\mathcal{H}$ be a Hilbert space with associated inner product $\langle \mathbf{x}, \mathbf{y} \rangle$, $\forall \mathbf{x}, \mathbf{y} \in \mathcal{H}$. The inner product induces the norm $\|\mathbf{x}\|^2 := \langle \mathbf{x}, \mathbf{x} \rangle$, $\forall \mathbf{x} \in \mathcal{H}$. Let $\mathcal{A} \subset \mathcal{H}$ be a compact set (the "set of atoms" or dictionary) and let $f \colon \mathcal{H} \to \mathbb{R}$ be convex and $L$-smooth ($L$-Lipschitz gradient in the finite dimensional case). If $\mathcal{H}$ is an infinite-dimensional Hilbert space, then $f$ is assumed to be *Fréchet differentiable*. The generalized MP algorithm studied in [32], presented in Algorithm 1, solves the following optimization problem:

$$\min_{\mathbf{x} \in \mathrm{lin}(\mathcal{A})} f(\mathbf{x}). \tag{1}$$

In each iteration, MP queries a linear minimization oracle (LMO) solving the following linear problem:

$$\mathrm{LMO}_{\mathcal{A}}(\mathbf{y}) := \arg\min_{\mathbf{z} \in \mathcal{A}} \langle \mathbf{y}, \mathbf{z} \rangle \tag{2}$$

for a given query $\mathbf{y} \in \mathcal{H}$. The MP update step minimizes a quadratic upper bound $g_{\mathbf{x}_t}(\mathbf{x}) = f(\mathbf{x}_t) + \langle \nabla f(\mathbf{x}_t), \mathbf{x} - \mathbf{x}_t \rangle + \frac{L}{2} \|\mathbf{x} - \mathbf{x}_t\|^2$ of $f$ at $\mathbf{x}_t$, where $L$ is an upper bound on the smoothness

constant of $f$ with respect to a chosen norm $\|\cdot\|$. Optimizing this norm problem instead of $f$ directly allows for substantial efficiency gains in the case of complicated $f$. For symmetric $\mathcal{A}$ and for $f(\mathbf{x}) = \frac{1}{2}\|\mathbf{y} - \mathbf{x}\|^2$, $\mathbf{y} \in \mathcal{H}$, Algorithm 1 recovers MP (Variant 0) [34] and OMP (Variant 1) [9, 49], see [32] for details.

---

**Algorithm 1** Norm-Corrective Generalized Matching Pursuit

---

1: **init** $\mathbf{x}_0 \in \mathrm{lin}(\mathcal{A})$, and $\mathcal{S} := \{\mathbf{x}_0\}$
2: **for** $t = 0 \ldots T$
3:     Find $\mathbf{z}_t := (\text{Approx-})\mathrm{LMO}_{\mathcal{A}}(\nabla f(\mathbf{x}_t))$
4:     $\mathcal{S} := \mathcal{S} \cup \{\mathbf{z}_t\}$
5:     Let $\mathbf{b} := \mathbf{x}_t - \frac{1}{L}\nabla f(\mathbf{x}_t)$
6:     Variant 0:
        Update $\mathbf{x}_{t+1} := \underset{\substack{\mathbf{z}:=\mathbf{x}_t + \gamma \mathbf{z}_t \\ \gamma \in \mathbb{R}}}{\arg\min} \|\mathbf{z} - \mathbf{b}\|^2$
7:     Variant 1:
        Update $\mathbf{x}_{t+1} := \underset{\mathbf{z} \in \mathrm{lin}(\mathcal{S})}{\arg\min} \|\mathbf{z} - \mathbf{b}\|^2$
8:     *Optional:* Correction of some/all atoms $\mathbf{z}_{0\ldots t}$
9: **end for**

---

**Approximate linear oracles.** Solving the LMO defined in (2) exactly is often hard in practice, in particular when applied to matrix (or tensor) factorization problems, while approximate versions can be much more efficient. Algorithm 1 allows for an *approximate* LMO. For given quality parameter $\delta \in (0, 1]$ and given direction $\mathbf{d} \in \mathcal{H}$, the approximate LMO for Algorithm 1 returns a vector $\tilde{\mathbf{z}} \in \mathcal{A}$ such that

$$\langle \mathbf{d}, \tilde{\mathbf{z}} \rangle \leq \delta \langle \mathbf{d}, \mathbf{z} \rangle, \qquad (3)$$

relative to $\mathbf{z} = \mathrm{LMO}_{\mathcal{A}}(\mathbf{d})$ being an exact solution.

**Discussion and limitations of MP.** The analysis of the convergence of Algorithm 1 in [32] critically relies on the assumption that the origin is in the relative interior of $\mathrm{conv}(\mathcal{A})$ with respect to its linear span. This assumption originates from the fact that the convergence of MP- and FW-type algorithms fundamentally depends on an *alignment assumption* of the search direction returned by the LMO (i.e., $\mathbf{z}_t$ in Algorithm 1) and the gradient of the objective at the current iteration (see *third premise* in [39]). Specifically, for Algorithm 1, the LMO is assumed to select a descent direction, i.e., $\langle \nabla f(\mathbf{x}_t), \mathbf{z}_t \rangle < 0$, so that the resulting weight (i.e., $\gamma$ for Variant 0) is always positive. In this spirit, Algorithm 1 is a natural candidate to minimize $f$ over the conic hull of $\mathcal{A}$. However, if the optimization domain is a cone, the alignment assumption does not hold as there may be non-stationary points $\mathbf{x}$ in the conic hull of $\mathcal{A}$ for which $\min_{\mathbf{z} \in \mathcal{A}} \langle \nabla f(\mathbf{x}), \mathbf{z} \rangle = 0$. Algorithm 1 is therefore not guaranteed to converge when applied to conic problems. The same issue arises for essentially all existing non-negative variants of MP, see, e.g., Alg. 2 in [38] and in Alg. 2 in [52]. We now present modifications corroborating this issue along with the resulting MP-type algorithms for conic problems and corresponding convergence guarantees.

## 3 Greedy Algorithms on Conic Hulls

The cone $\mathrm{cone}(\mathcal{A} - \mathbf{y})$ tangent to the convex set $\mathrm{conv}(\mathcal{A})$ at a point $\mathbf{y}$ is formed by the half-lines emanating from $\mathbf{y}$ and intersecting $\mathrm{conv}(\mathcal{A})$ in at least one point distinct from $\mathbf{y}$. Without loss of generality we consider $\mathbf{0} \in \mathcal{A}$ and assume the set $\mathrm{cone}(\mathcal{A})$ (i.e., $\mathbf{y} = \mathbf{0}$) to be closed. If $\mathcal{A}$ is finite the cone constraint can be written as $\mathrm{cone}(\mathcal{A}) := \{\mathbf{x} : \mathbf{x} = \sum_{i=1}^{|\mathcal{A}|} \alpha_i \mathbf{a}_i \ \text{s.t.} \ \mathbf{a}_i \in \mathcal{A}, \ \alpha_i \geq 0 \ \forall i\}$. We consider conic optimization problems of the form:

$$\min_{\mathbf{x} \in \mathrm{cone}(\mathcal{A})} f(\mathbf{x}). \qquad (4)$$

Note that if the set $\mathcal{A}$ is symmetric or if the origin is in the relative interior of $\mathrm{conv}(\mathcal{A})$ w.r.t. its linear span then $\mathrm{cone}(\mathcal{A}) = \mathrm{lin}(\mathcal{A})$. We will show later how our results recover known MP rates when the origin is in the relative interior of $\mathrm{conv}(\mathcal{A})$.

As a first algorithm to solve problems of the form (4), we present the Non-Negative Generalized Matching Pursuit (NNMP) in Algorithm 2 which is an extension of MP to general $f$ and non-negative weights.

**Discussion:** Algorithm 2 differs from Algorithm 1 (Variant 0) in line 4, adding the iteration-dependent atom $-\frac{\mathbf{x}_t}{\|\mathbf{x}_t\|_{\mathcal{A}}}$ to the set of possible search directions[1]. We use the atomic norm for the

**Algorithm 2** Non-Negative Matching Pursuit

1: **init** $\mathbf{x}_0 = \mathbf{0} \in \mathcal{A}$
2: **for** $t = 0 \ldots T$
3:     Find $\bar{\mathbf{z}}_t := (\text{Approx-})\text{LMO}_{\mathcal{A}}(\nabla f(\mathbf{x}_t))$
4:     $\mathbf{z}_t = \arg\min_{\mathbf{z} \in \left\{\bar{\mathbf{z}}_t, \frac{-\mathbf{x}_t}{\|\mathbf{x}_t\|_{\mathcal{A}}}\right\}} \langle \nabla f(\mathbf{x}_t), \mathbf{z} \rangle$
5:     $\gamma := \frac{\langle -\nabla f(\mathbf{x}_t), \mathbf{z}_t \rangle}{L \|\mathbf{z}_t\|^2}$
6:     Update $\mathbf{x}_{t+1} := \mathbf{x}_t + \gamma \mathbf{z}_t$
7: **end for**

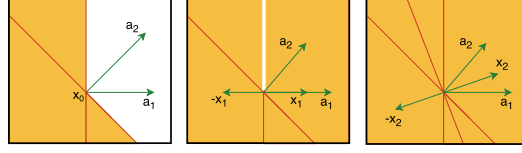

Figure 1: Two dimensional example for $T_{\mathcal{A}}(\mathbf{x}_t)$ where $\mathcal{A} = \{\mathbf{a}_1, \mathbf{a}_2\}$, for three different iterates $\mathbf{x}_0$, $\mathbf{x}_1$ and $\mathbf{x}_2$. The shaded area corresponds to $T_{\mathcal{A}}(\mathbf{x}_t)$ and the white area to $\text{lin}(\mathcal{A}) \setminus T_{\mathcal{A}}(\mathbf{x}_t)$.

normalization because it yields the best constant in the convergence rate. In practice, one can replace it with the Euclidean norm, which is often much less expensive to compute. This iteration-dependent additional search direction allows to reduce the weights of the atoms that were previously selected, thus admitting the algorithm to "move back" towards the origin while maintaining the cone constraint. This idea is informally explained here and formally studied in Section 4.1.

Recall the alignment assumption of the search direction and the gradient of the objective at the current iterate discussed in Section 2 (see also [39]). Algorithm 2 obeys this assumption. The intuition behind this is the following. Whenever $\mathbf{x}_t$ is not a minimizer of (4) and $\min_{\mathbf{z} \in \mathcal{A}} \langle \nabla f(\mathbf{x}_t), \mathbf{z} \rangle = 0$, the vector $-\frac{\mathbf{x}_t}{\|\mathbf{x}_t\|_{\mathcal{A}}}$ is aligned with $\nabla f(\mathbf{x}_t)$ (i.e., $\langle \nabla f(\mathbf{x}_t), -\frac{\mathbf{x}_t}{\|\mathbf{x}_t\|_{\mathcal{A}}} \rangle < 0$), preventing the algorithm from stopping at a suboptimal iterate. To make this intuition more formal, let us define the set of feasible descent directions of Algorithm 2 at a point $\mathbf{x} \in \text{cone}(\mathcal{A})$ as:

$$T_{\mathcal{A}}(\mathbf{x}) := \left\{ \mathbf{d} \in \mathcal{H} \colon \exists \mathbf{z} \in \mathcal{A} \cup \left\{ -\frac{\mathbf{x}}{\|\mathbf{x}\|_{\mathcal{A}}} \right\} \text{ s.t. } \langle \mathbf{d}, \mathbf{z} \rangle < 0 \right\}. \tag{5}$$

If at some iteration $t = 0, 1, \ldots$ the gradient $\nabla f(\mathbf{x}_t)$ is not in $T_{\mathcal{A}}(\mathbf{x}_t)$ Algorithm 2 terminates as $\min_{\mathbf{z} \in \mathcal{A}} \langle \mathbf{d}, \mathbf{z} \rangle = 0$ and $\langle \mathbf{d}, -\mathbf{x}_t \rangle \geq 0$ (which yields $\mathbf{z}_t = 0$). Even though, in general, not every direction in $\mathcal{H}$ is a feasible descent direction, $\nabla f(\mathbf{x}_t) \notin T_{\mathcal{A}}$ only occurs if $\mathbf{x}_t$ is a constrained minimum of Equation 4:

**Lemma 1.** *If* $\tilde{\mathbf{x}} \in \text{cone}(\mathcal{A})$ *and* $\nabla f(\tilde{\mathbf{x}}) \notin T_{\mathcal{A}}$ *then* $\tilde{\mathbf{x}}$ *is a solution to* $\min_{\mathbf{x} \in \text{cone}(\mathcal{A})} f(\mathbf{x})$.

Initializing Algorithm 2 with $\mathbf{x}_0 = \mathbf{0}$ guarantees that the iterates $\mathbf{x}_t$ always remain inside $\text{cone}(\mathcal{A})$ even though this is not enforced explicitly (by convexity of $f$, see proof of Theorem 2 in Appendix D for details).

**Limitations of Algorithm 2:** Let us call *active* the atoms which have nonzero weights in the representation of $\mathbf{x}_t = \sum_{i=0}^{t-1} \alpha_i \mathbf{z}_i$ computed by Algorithm 2. Formally, the set of active atoms is defined as $\mathcal{S} := \{\mathbf{z}_i \colon \alpha_i > 0, i = 0, 1, \ldots, t - 1\}$. The main drawback of Algorithm 2 is that when the direction $-\frac{\mathbf{x}_t}{\|\mathbf{x}_t\|_{\mathcal{A}}}$ is selected, the weight of *all* active atoms is reduced. This can lead to the algorithm alternately selecting $-\frac{\mathbf{x}_t}{\|\mathbf{x}_t\|_{\mathcal{A}}}$ and an atom from $\mathcal{A}$, thereby slowing down convergence in a similar manner as the *zig-zagging* phenomenon well-known in the Frank-Wolfe framework [28]. In order to achieve faster convergence we introduce the corrective variants of Algorithm 2.

### 3.1 Corrective Variants

To achieve faster (linear) convergence (see Section 4.2) we introduce variants of Algorithm 2, termed Away-steps MP (AMP) and Pairwise MP (PWMP), presented in Algorithm 3. Here, inspired by the away-steps and pairwise variants of FW [12, 28], instead of reducing the weights of the active atoms uniformly as in Algorithm 2, the LMO is queried a second time on the active set $\mathcal{S}$ to identify the direction of steepest ascent in $\mathcal{S}$. This allows, at each iteration, to reduce the weight of a previously selected atom (AMP) or swap weight between atoms (PWMP). This selective "reduction" or "swap of weight" helps to avoid the zig-zagging phenomenon which prevent Algorithm 2 from converging linearly.

At each iteration, Algorithm 3 updates the weights of $\mathbf{z}_t$ and $\mathbf{v}_t$ as $\alpha_{\mathbf{z}_t} = \alpha_{\mathbf{z}_t} + \gamma$ and $\alpha_{\mathbf{v}_t} = \alpha_{\mathbf{v}_t} - \gamma$, respectively. To ensure that $\mathbf{x}_{t+1} \in \text{cone}(\mathcal{A})$, $\gamma$ has to be clipped according to the weight which is currently on $\mathbf{v}_t$, i.e., $\gamma_{\max} = \alpha_{\mathbf{v}_t}$. If $\gamma = \gamma_{\max}$, we set $\alpha_{\mathbf{v}_t} = 0$ and remove $\mathbf{v}_t$ from $\mathcal{S}$ as the atom $\mathbf{v}_t$ is no longer active. If $\mathbf{d}_t \in \mathcal{A}$ (i.e., we take a regular MP step and not an away step), the line search is unconstrained (i.e., $\gamma_{\max} = \infty$).

For both algorithm variants, the second LMO query increases the computational complexity. Note that an exact search on $\mathcal{S}$ is feasible in practice as $|\mathcal{S}|$ has at most $t$ elements at iteration $t$.

Taking an additional computational burden allows to update the weights of all active atoms in the spirit of OMP. This approach is implemented in the Fully Corrective MP (FCMP), Algorithm 4.

| **Algorithm 3** Away-steps (AMP) and Pairwise (PWMP) Non-Negative Matching Pursuit | **Algorithm 4** Fully Corrective Non-Negative Matching Pursuit (FCMP) |
|---|---|
| 1: **init** $\mathbf{x}_0 = \mathbf{0} \in \mathcal{A}$, and $\mathcal{S} := \{\mathbf{x}_0\}$<br>2: **for** $t = 0 \dots T$<br>3:     Find $\mathbf{z}_t := (\text{Approx-})\text{LMO}_{\mathcal{A}}(\nabla f(\mathbf{x}_t))$<br>4:     Find $\mathbf{v}_t := (\text{Approx-})\text{LMO}_{\mathcal{S}}(-\nabla f(\mathbf{x}_t))$<br>5:     $\mathcal{S} = \mathcal{S} \cup \mathbf{z}_t$<br>6:     *AMP:* $\mathbf{d}_t = \arg\min_{\mathbf{d} \in \{\mathbf{z}_t, -\mathbf{v}_t\}} \langle \nabla f(\mathbf{x}_t), \mathbf{d} \rangle$<br>7:     *PWMP:* $\mathbf{d}_t = \mathbf{z}_t - \mathbf{v}_t$<br>8:     $\gamma := \min\left\{\frac{\langle -\nabla f(\mathbf{x}_t), \mathbf{d}_t \rangle}{L \|\mathbf{d}_t\|^2}, \gamma_{\max}\right\}$<br>        ($\gamma_{\max}$ see text)<br>9:     Update $\alpha_{\mathbf{z}_t}$, $\alpha_{\mathbf{v}_t}$ and $\mathcal{S}$ according to $\gamma$<br>        ($\gamma$ see text)<br>10:    Update $\mathbf{x}_{t+1} := \mathbf{x}_t + \gamma \mathbf{d}_t$<br>11: **end for** | 1: **init** $\mathbf{x}_0 = \mathbf{0} \in \mathcal{A}, \mathcal{S} = \{\mathbf{x}_0\}$<br>2: **for** $t = 0 \dots T$<br>3:     Find $\mathbf{z}_t := (\text{Approx-})\text{LMO}_{\mathcal{A}}(\nabla f(\mathbf{x}_t))$<br>4:     $\mathcal{S} := \mathcal{S} \cup \{\mathbf{z}_t\}$<br>5:     *Variant 0:*<br>      $\mathbf{x}_{t+1} = \arg\min_{\mathbf{x} \in \text{cone}(\mathcal{S})} \|\mathbf{x} - (\mathbf{x}_t - \frac{1}{L}\nabla f(\mathbf{x}_t))\|^2$<br>6:     *Variant 1:*<br>      $\mathbf{x}_{t+1} = \arg\min_{\mathbf{x} \in \text{cone}(\mathcal{S})} f(\mathbf{x})$<br>7:     Remove atoms with zero weights from $\mathcal{S}$<br>8: **end for** |

At each iteration, Algorithm 4 maintains the set of active atoms $\mathcal{S}$ by adding $\mathbf{z}_t$ and removing atoms with zero weights after the update. In Variant 0, the algorithm minimizes the quadratic upper bound $g_{\mathbf{x}_t}(\mathbf{x})$ on $f$ at $\mathbf{x}_t$ (see Section 2) imitating a gradient descent step with projection onto a "varying" target, i.e., $\text{cone}(\mathcal{S})$. In Variant 1, the original objective $f$ is minimized over $\text{cone}(\mathcal{S})$ at each iteration, which is in general more efficient than minimizing $f$ over $\text{cone}(\mathcal{A})$ using a generic solver for cone constrained problems. For $f(\mathbf{x}) = \frac{1}{2}\|\mathbf{y} - \mathbf{x}\|^2$, $\mathbf{y} \in \mathcal{H}$, Variant 1 recovers Algorithm 1 in [52] and the OMP variant in [5] which both only apply to this specific objective $f$.

### 3.2 Computational Complexity

We briefly discuss the computational complexity of the algorithms we introduced. For $\mathcal{H} = \mathbb{R}^d$, sums and inner products have cost $O(d)$. Let us assume that each call of the LMO has cost $C$ on the set $\mathcal{A}$ and $O(td)$ on $\mathcal{S}$. The variants 0 and 1 of FCMP solve a cone problem at each iteration with cost $h_0$ and $h_1$, respectively. In general, $h_0$ can be much smaller than $h_1$. In Table 1

| algorithm | cost per iteration | convergence | $k(t)$ |
|---|---|---|---|
| NNMP | $C + O(d)$ | $O(1/t)$ | - |
| PWMP | $C + O(d + td)$ | $O\left(e^{-\beta k(t)}\right)$ | $\frac{t}{3|\mathcal{A}|!+1}$ |
| AMP | $C + O(d + td)$ | $O\left(e^{-\frac{\beta}{2}k(t)}\right)$ | $t/2$ |
| FCMP v. 0 | $C + O(d) + h_0$ | $O\left(e^{-\beta k(t)}\right)$ | $\frac{t}{3|\mathcal{A}|!+1}$ |
| FCMP v. 1 | $C + O(d) + h_1$ | $O\left(e^{-\beta k(t)}\right)$ | $t$ |

Table 1: Computational complexity versus convergence rate (see Section 4) for strongly convex objectives

we report the cost per iteration for every algorithm along with the asymptotic convergence rates derived in Section 4.

## 4 Convergence Rates

In this section, we present convergence guarantees for Algorithms 2, 3, and 4. All proofs are deferred to the Appendix in the supplementary material. We write $\mathbf{x}^\star \in \arg\min_{\mathbf{x} \in \text{cone}(\mathcal{A})} f(\mathbf{x})$ for an optimal solution. Our rates will depend on the atomic norm of the solution and the iterates of the respective algorithm variant:

$$\rho = \max\left\{\|\mathbf{x}^\star\|_{\mathcal{A}}, \|\mathbf{x}_0\|_{\mathcal{A}} \dots, \|\mathbf{x}_T\|_{\mathcal{A}}\right\}. \tag{6}$$

If the optimum is not unique, we consider $\mathbf{x}^\star$ to be one of largest atomic norm. A more intuitive and looser notion is to simply upper-bound $\rho$ by the diameter of the level set of the initial iterate $\mathbf{x}_0$ measured by the atomic norm. Then, boundedness follows since the presented method is a descent method (due to Lemma 1 and line search on the quadratic upper bound, each iteration strictly

decreases the objective and our method stops only at the optimum). This justifies the statement $f(\mathbf{x}_t) \leq f(\mathbf{x}_0)$. Hence, $\rho$ must be bounded for any sequence of iterates produced by the algorithm, and the convergence rates presented in this section are valid as $T$ goes to infinity. A similar notion to measure the convergence of MP was established in [32]. All of our algorithms and rates can be made *affine invariant*. We defer this discussion to Appendix B.

## 4.1 Sublinear Convergence

We now present the convergence results for the non-negative and Fully-Corrective Matching Pursuit algorithms. Sublinear convergence of Algorithm 3 is addressed in Theorem 3.

**Theorem 2.** *Let $\mathcal{A} \subset \mathcal{H}$ be a bounded set with $\mathbf{0} \in \mathcal{A}$, $\rho := \max\{\|\mathbf{x}^\star\|_\mathcal{A}, \|\mathbf{x}_0\|_\mathcal{A}, \ldots, \|\mathbf{x}_T\|_\mathcal{A}, \}$ and $f$ be $L$-smooth over $\rho \operatorname{conv}(\mathcal{A} \cup -\mathcal{A})$. Then, Algorithms 2 and 4 converge for $t \geq 0$ as*

$$f(\mathbf{x}_t) - f(\mathbf{x}^\star) \leq \frac{4\left(\frac{2}{\delta}L\rho^2 \operatorname{radius}(\mathcal{A})^2 + \varepsilon_0\right)}{\delta t + 4},$$

*where $\delta \in (0, 1]$ is the relative accuracy parameter of the employed approximate* LMO *(see Equation (3)).*

**Relation to FW rates.** By rescaling $\mathcal{A}$ by a large enough factor $\tau > 0$, FW with $\tau\mathcal{A}$ as atom set could in principle be used to solve (4). In fact, for large enough $\tau$, only the constraints of (4) become active when minimizing $f$ over $\operatorname{conv}(\tau\mathcal{A})$. The sublinear convergence rate obtained with this approach is up to constants identical to that in Theorem 2 for our MP variants, see [21]. However, as the correct scaling is unknown, one has to either take the risk of choosing $\tau$ too small and hence failing to recover an optimal solution of (4), or to rely on too large $\tau$ which can result in slow convergence. In contrast, knowledge of $\rho$ is not required to run our MP variants.

**Relation to MP rates.** If $\mathcal{A}$ is symmetric, we have that $\operatorname{lin}(\mathcal{A}) = \operatorname{cone}(\mathcal{A})$ and it is easy to show that the additional direction $-\frac{\mathbf{x}_t}{\|\mathbf{x}_t\|}$ in Algorithm 2 is never selected. Therefore, Algorithm 2 becomes equivalent to Variant 0 of Algorithm 1, while Variant 1 of Algorithm 1 is equivalent to Variant 0 of Algorithm 4. The rate specified in Theorem 2 hence generalizes the sublinear rate in [32, Theorem 2] for symmetric $\mathcal{A}$.

## 4.2 Linear Convergence

We start by recalling some of the geometric complexity quantities that were introduced in the context of FW and are adapted here to the optimization problem we aim to solve (minimization over $\operatorname{cone}(\mathcal{A})$ instead of $\operatorname{conv}(\mathcal{A})$).

**Directional Width.** The directional width of a set $\mathcal{A}$ w.r.t. a direction $\mathbf{r} \in \mathcal{H}$ is defined as:

$$dirW(\mathcal{A}, \mathbf{r}) := \max_{\mathbf{s}, \mathbf{v} \in \mathcal{A}} \left\langle \frac{\mathbf{r}}{\|\mathbf{r}\|}, \mathbf{s} - \mathbf{v} \right\rangle \tag{7}$$

**Pyramidal Directional Width** [28]**.** The Pyramidal Directional Width of a set $\mathcal{A}$ with respect to a direction $\mathbf{r}$ and a reference point $\mathbf{x} \in \operatorname{conv}(\mathcal{A})$ is defined as:

$$PdirW(\mathcal{A}, \mathbf{r}, \mathbf{x}) := \min_{\mathcal{S} \in \mathcal{S}_\mathbf{x}} dirW(\mathcal{S} \cup \{\mathbf{s}(\mathcal{A}, \mathbf{r})\}, \mathbf{r}), \tag{8}$$

where $\mathcal{S}_\mathbf{x} := \{\mathcal{S} \mid \mathcal{S} \subset \mathcal{A} \text{ and } \mathbf{x} \text{ is a proper convex combination of all the elements in } \mathcal{S}\}$ and $\mathbf{s}(\mathcal{A}, \mathbf{r}) := \max_{\mathbf{s} \in \mathcal{A}} \langle \frac{\mathbf{r}}{\|\mathbf{r}\|}, \mathbf{s} \rangle$.

Inspired by the notion of pyramidal width in [28], which is the minimal pyramidal directional width computed over the set of feasible directions, we now define the cone width of a set $\mathcal{A}$ where only the generating faces (g-faces) of $\operatorname{cone}(\mathcal{A})$ (instead of the faces of $\operatorname{conv}(\mathcal{A})$) are considered. Before doing so we introduce the notions of *face*, *generating face*, and *feasible direction*.

**Face of a convex set.** Let us consider a set $\mathcal{K}$ with a $k-$dimensional affine hull along with a point $\mathbf{x} \in \mathcal{K}$. Then, $\mathcal{K}$ is a $k-$dimensional face of $\operatorname{conv}(\mathcal{A})$ if $\mathcal{K} = \operatorname{conv}(\mathcal{A}) \cap \{\mathbf{y} : \langle \mathbf{r}, \mathbf{y} - \mathbf{x} \rangle = 0\}$ for some normal vector $\mathbf{r}$ and $\operatorname{conv}(A)$ is contained in the half-space determined by $\mathbf{r}$, i.e., $\langle \mathbf{r}, \mathbf{y} - \mathbf{x} \rangle \leq 0, \forall \mathbf{y} \in \operatorname{conv}(\mathcal{A})$. Intuitively, given a set $\operatorname{conv}(\mathcal{A})$ one can think of $\operatorname{conv}(\mathcal{A})$ being a $\dim(\operatorname{conv}(\mathcal{A}))-$dimensional face of itself, an edge on the border of the set a 1-dimensional face and a vertex a 0-dimensional face.

**Face of a cone and g-faces.** Similarly, a $k-$dimensional face of a cone is an open and unbounded set $\text{cone}(\mathcal{A}) \cap \{\mathbf{y}\colon \langle \mathbf{r}, \mathbf{y} - \mathbf{x} \rangle = 0\}$ for some normal vector $\mathbf{r}$ and $\text{cone}(A)$ is contained in the half space determined by $\mathbf{r}$. We can define the generating faces of a cone as:

$$\text{g-faces}(\text{cone}(\mathcal{A})) := \{\mathcal{B} \cap \text{conv}(\mathcal{A}) \colon \mathcal{B} \in \text{faces}(\text{cone}(\mathcal{A}))\}.$$

Note that $\text{g-faces}(\text{cone}(\mathcal{A})) \subset \text{faces}(\text{conv}(\mathcal{A}))$ and $\text{conv}(\mathcal{A}) \in \text{g-faces}(\text{cone}(\mathcal{A}))$. Furthermore, for each $\mathcal{K} \in \text{g-faces}(\text{cone}(\mathcal{A}))$, $\text{cone}(\mathcal{K})$ is a $k-$dimensional face of $\text{cone}(\mathcal{A})$.

We now introduce the notion of **feasible directions**. A direction $\mathbf{d}$ is feasible from $\mathbf{x} \in \text{cone}(\mathcal{A})$ if it points inwards $\text{cone}(\mathcal{A})$, i.e., if $\exists \varepsilon > 0$ s.t. $\mathbf{x} + \varepsilon \mathbf{d} \in \text{cone}(\mathcal{A})$. Since a face of the cone is itself a cone, if a direction is feasible from $\mathbf{x} \in \text{cone}(\mathcal{K}) \setminus \mathbf{0}$, it is feasible from every positive rescaling of $\mathbf{x}$. We therefore can consider only the feasible directions on the generating faces (which are closed and bounded sets). Finally, we define the cone width of $\mathcal{A}$.

**Cone Width.**

$$\text{CWidth}(\mathcal{A}) := \min_{\substack{\mathcal{K} \in \text{g-faces}(\text{cone}(\mathcal{A})) \\ \mathbf{x} \in \mathcal{K} \\ \mathbf{r} \in \text{cone}(\mathcal{K}-\mathbf{x}) \setminus \{\mathbf{0}\}}} PdirW(\mathcal{K} \cap \mathcal{A}, \mathbf{r}, \mathbf{x}) \tag{9}$$

We are now ready to show the linear convergence of Algorithms 3 and 4.

**Theorem 3.** *Let $\mathcal{A} \subset \mathcal{H}$ be a bounded set with $\mathbf{0} \in \mathcal{A}$ and let the objective function $f\colon \mathcal{H} \to \mathbb{R}$ be both L-smooth and $\mu$-strongly convex over $\rho \, \text{conv}(\mathcal{A} \cup -\mathcal{A})$. Then, the suboptimality of the iterates of Algorithms 3 and 4 decreases geometrically at each step in which $\gamma < \alpha_{\mathbf{v}_t}$ (henceforth referred to as "good steps") as:*

$$\varepsilon_{t+1} \leq (1 - \beta) \, \varepsilon_t, \tag{10}$$

*where $\beta := \delta^2 \frac{\mu \, \text{CWidth}(\mathcal{A})^2}{L \, \text{diam}(\mathcal{A})^2} \in (0,1]$, $\varepsilon_t := f(\mathbf{x}_t) - f(\mathbf{x}^\star)$ is the suboptimality at step $t$ and $\delta \in (0,1]$ is the relative accuracy parameter of the employed approximate LMO (3). For AMP (Algorithm 3), $\beta^{\text{AMP}} = \beta/2$. If $\mu = 0$ Algorithm 3 converges with rate $O(1/k(t))$ where $k(t)$ is the number of "good steps" up to iteration t.*

**Discussion.** To obtain a linear convergence rate, one needs to upper-bound the number of "bad steps" $t - k(t)$ (i.e., steps with $\gamma \geq \alpha_{\mathbf{v}_t}$). We have that $k(t) = t$ for Variant 1 of FCMP (Algorithm 4), $k(t) \geq t/2$ for AMP (Algorithm 3) and $k(t) \geq t/(3|\mathcal{A}|! + 1)$ for PWMP (Algorithm 3) and Variant 0 of FCMP (Algorithm 4). This yields a global linear convergence rate of $\varepsilon_t \leq \varepsilon_0 \exp(-\beta k(t))$. The bound for PWMP is very loose and only meaningful for finite sets $\mathcal{A}$. However, it can be observed in the experiments in the supplementary material (Appendix A) that only a very small fraction of iterations result in bad PWMP steps in practice. Further note that Variant 1 of FCMP (Algorithm 4) does not produce bad steps. Also note that the bounds on the number of good steps given above are the same as for the corresponding FW variants and are obtained using the same (purely combinatorial) arguments as in [28].

**Relation to previous MP rates.** The linear convergence of the generalized (not non-negative) MP variants studied in [32] crucially depends on the geometry of the set which is characterized by the Minimal Directional Width $\text{mDW}(\mathcal{A})$:

$$\text{mDW}(\mathcal{A}) := \min_{\substack{\mathbf{d} \in \text{lin}(\mathcal{A}) \\ \mathbf{d} \neq \mathbf{0}}} \max_{\mathbf{z} \in \mathcal{A}} \langle \frac{\mathbf{d}}{\|\mathbf{d}\|}, \mathbf{z} \rangle. \tag{11}$$

The following Lemma relates the Cone Width with the minimal directional width.

**Lemma 4.** *If the origin is in the relative interior of $\text{conv}(\mathcal{A})$ with respect to its linear span, then $\text{cone}(\mathcal{A}) = \text{lin}(\mathcal{A})$ and $\text{CWidth}(\mathcal{A}) = \text{mDW}(\mathcal{A})$.*

Now, if the set $\mathcal{A}$ is symmetric or, more generally, if $\text{cone}(\mathcal{A})$ spans the linear space $\text{lin}(\mathcal{A})$ (which implies that the origin is in the relative interior of $\text{conv}(\mathcal{A})$), there are no bad steps. Hence, by Lemma 4, the linear rate obtained in Theorem 3 for non-negative MP variants generalizes the one presented in [32, Theorem 7] for generalized MP variants.

**Relation to FW rates.** Optimization over conic hulls with non-negative MP is more similar to FW than to MP itself in the following sense. For MP, every direction in $\mathrm{lin}(\mathcal{A})$ allows for unconstrained steps, from any iterate $\mathbf{x}_t$. In contrast, for our non-negative MPs, while some directions allow for unconstrained steps from some iterate $\mathbf{x}_t$, others are constrained, thereby leading to the dependence of the linear convergence rate on the cone width, a geometric constant which is very similar in spirit to the Pyramidal Width appearing in the linear convergence bound in [28] for FW. Furthermore, as for Algorithm 3, the linear rate of Away-steps and Pairwise FW holds only for good steps. We finally relate the cone width with the Pyramidal Width [28]. The Pyramidal Width is defined as

$$\mathrm{PWidth}(\mathcal{A}) := \min_{\substack{\mathcal{K}\in\mathrm{faces}(\mathrm{conv}(\mathcal{A}))\\ \mathbf{x}\in\mathcal{K}\\ \mathbf{r}\in\mathrm{cone}(\mathcal{K}-\mathbf{x})\setminus\{\mathbf{0}\}}} PdirW(\mathcal{K}\cap\mathcal{A}, \mathbf{r}, \mathbf{x}).$$

We have $\mathrm{CWidth}(\mathcal{A}) \geq \mathrm{PWidth}(\mathcal{A})$ as the minimization in the definition (9) of $\mathrm{CWidth}(\mathcal{A})$ is only over the subset g-faces$(\mathrm{cone}(\mathcal{A}))$ of faces$(\mathrm{conv}(\mathcal{A}))$. As a consequence, the decrease per iteration characterized in Theorem 3 is larger than what one could obtain with FW on the rescaled convex set $\tau\mathcal{A}$ (see Section 4.1 for details about the rescaling). Furthermore, the decrease characterized in [28] scales as $1/\tau^2$ due to the dependence on $1/\mathrm{diam}(\mathrm{conv}(\mathcal{A}))^2$.

# 5 Related Work

The line of recent works by [44, 46, 47, 48, 37, 32] targets the generalization of MP from the least-squares objective to general smooth objectives and derives corresponding convergence rates (see [32] for a more in-depth discussion). However, only little prior work targets MP variants with non-negativity constraint [5, 38, 52]. In particular, the least-squares objective was addressed and no rigorous convergence analysis was carried out. [5, 52] proposed an algorithm equivalent to our Algorithm 4 for the least-squares case. More specifically, [52] then developed an acceleration heuristic, whereas [5] derived a coherence-based recovery guarantee for sparse linear combinations of atoms. Apart from MP-type algorithms, there is a large variety of non-negative least-squares algorithms, e.g., [30], in particular also for matrix and tensor spaces. The gold standard in factorization problems is projected gradient descent with alternating minimization, see [43, 4, 45, 23]. Other related works are [40], which is concerned with the feasibility problem on symmetric cones, and [19], which introduces a norm-regularized variant of problem (4) and solves it using FW on a rescaled convex set. To the best of our knowledge, in the context of MP-type algorithms, we are the first to combine general convex objectives with conic constraints and to derive corresponding convergence guarantees.

**Boosting:** In an earlier line of work, a flavor of the generalized MP became popular in the context of boosting, see [35]. The literature on boosting is vast, we refer to [42, 35, 7] for a general overview. Taking the optimization perspective given in [42], boosting is an iterative greedy algorithm minimizing a (strongly) convex objective over the linear span of a possibly infinite set called hypothesis class. The convergence analysis crucially relies on the assumption of the origin being in the relative interior of the hypothesis class, see Theorem 1 in [17]. Indeed, Algorithm 5.2 of [35] might not converge if the [39] alignment assumption is violated. Here, we managed to relax this assumption while preserving essentially the same asymptotic rates in [35, 17]. Our work is therefore also relevant in the context of (non-negative) boosting.

# 6 Illustrative Experiments

We illustrate the performance of the presented algorithms on three different exemplary tasks, showing that our algorithms are competitive with established baselines across a wide range of objective functions, domains, and data sets while not being specifically tailored to any of these tasks (see Section 3.2 for a discussion of the computational complexity of the algorithms). Additional experiments targeting KL divergence NMF, non-negative tensor factorization, and hyperspectral image unmixing can be found in the appendix.

**Synthetic data.** We consider minimizing the least squares objective on the conic hull of 100 unit-norm vectors sampled at random in the first orthant of $\mathbb{R}^{50}$. We compare the convergence of Algorithms 2, 3, and 4 with the Fast Non-Negative MP (FNNOMP) of [52], and Variant 3 (line-search) of the FW algorithm in [32] on the atom set rescaled by $\tau = 10\|\mathbf{y}\|$ (see Section 4.1), observing linear convergence for our corrective variants.

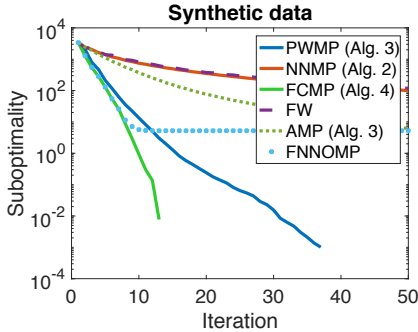

Figure 2: Synthetic data experiment.

Figure 2 shows the suboptimality $\varepsilon_t$, averaged over 20 realizations of $\mathcal{A}$ and $\mathbf{y}$, as a function of the iteration $t$. As expected, FCMP achieves fastest convergence followed by PWMP, AMP and NNMP. The FNNOMP gets stuck instead. Indeed, [52] only show that the algorithm terminates and not its convergence.

**Non-negative matrix factorization.** The second task consists of decomposing a given matrix into the product of two non-negative matrices as in Equation (1) of [20]. We consider the intersection of the positive semidefinite cone and the positive orthant. We parametrize the set $\mathcal{A}$ as the set of matrices obtained as an outer product of vectors from $\mathcal{A}_1 = \{\mathbf{z} \in \mathbb{R}^k : \mathbf{z}_i \geq 0 \ \forall \, i\}$ and $\mathcal{A}_2 = \{\mathbf{z} \in \mathbb{R}^d : \mathbf{z}_i \geq 0 \ \forall \, i\}$. The LMO is approximated using a truncated power method [55], and we perform atom correction with greedy coordinate descent see, e.g., [29, 18], to obtain a better objective value while maintaining the same (small) number of atoms. We consider three different datasets: The Reuters Corpus[2], the CBCL face dataset[3] and the KNIX dataset[4]. The subsample of the Reuters corpus we used is a term frequency matrix of 7,769 documents and 26,001 words. The CBCL face dataset is composed of 2,492 images of 361 pixels each, arranged into a matrix. The KNIX dataset contains 24 MRI slices of a knee, arranged in a matrix of size $262,144 \times 24$. Pixels are divided by their overall mean intensity. For interpretability reasons, there is interest to decompose MRI data into non-negative factorizations [25]. We compare PWMP and FCMP against the multiplicative (mult) and the alternating (als) algorithm of [4], and the greedy coordinate descent (GCD) of [20]. Since the Reuters corpus is much larger than the CBCL and the KNIX dataset we only used the GCD for which a fast implementation in C is available. We report the objective value for fixed values of the rank in Table 2, showing that FCMP outperform all the baselines across all the datasets. PWMP achieves smallest error on the Reuters corpus.

**Non-negative garrote.** We consider the non-negative garrote which is a common approach to model order selection [6]. We evaluate NNMP, PWMP, and FCMP in the experiment described in [33], where the non-negative garrote is used to perform model order selection for logistic regression (i.e., for a non-quadratic objective function). We evaluated training and test accuracy on 100 random splits of the sonar dataset from the UCI machine learning repository. In Table 3 we compare the median classification accuracy of our algorithms with that of the cyclic coordinate descent algorithm (NNG) from [33].

| algorithm | Reuters $K = 10$ | CBCL $K = 10$ | CBCL $K = 50$ | KNIX $K = 10$ |
|---|---|---|---|---|
| mult | - | 2.4241e3 | 1.1405e3 | 2.4471e03 |
| als | - | 2.73e3 | 3.84e3 | 2.7292e03 |
| GCD | 5.9799e5 | 2.2372e3 | 806 | 2.2372e03 |
| **PWMP** | **5.9591e5** | 2.2494e3 | 789.901 | 2.2494e03 |
| **FCMP** | 5.9762e5 | **2.2364e3** | **786.15** | **2.2364e03** |

Table 2: Objective value for least-squares non-negative matrix factorization with rank $K$.

| | training accuracy | test accuracy |
|---|---|---|
| **NNMP** | $0.8345 \pm 0.0242$ | $\mathbf{0.7419} \pm 0.0389$ |
| **PWMP** | $\mathbf{0.8379} \pm 0.0240$ | $\mathbf{0.7419} \pm 0.0392$ |
| **FCMP** | $0.8345 \pm 0.0238$ | $\mathbf{0.7419} \pm 0.0403$ |
| **NNG** | $0.8069 \pm 0.0518$ | $0.7258 \pm 0.0602$ |

Table 3: Logistic Regression with non-negative Garrote, median $\pm$ std. dev.

## 7 Conclusion

In this paper, we considered greedy algorithms for optimization over the convex cone, parametrized as the *conic hull* of a generic atom set. We presented a novel formulation of NNMP along with a comprehensive convergence analysis. Furthermore, we introduced corrective variants with linear convergence guarantees, and verified this convergence rate in numerical applications. We believe that the generality of our novel analysis will be useful to design new, fast algorithms with convergence guarantees, and to study convergence of existing heuristics, in particular in the context of non-negative matrix and tensor factorization.

## Footnotes

[1] This additional direction makes sense only if $\mathbf{x}_t \neq \mathbf{0}$. Therefore, we set $-\frac{\mathbf{x}_t}{\|\mathbf{x}_t\|_{\mathcal{A}}} = 0$ if $\mathbf{x}_t = \mathbf{0}$, i.e., no direction is added.

[2]http://www.nltk.org/book/ch02.html

[3]http://cbcl.mit.edu/software-datasets/FaceData2.html

[4]http://www.osirix-viewer.com/resources/dicom-image-library/

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
