[Supplementary Material]

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

$$CWidth(A) = \sin\frac{\theta}{2}$$

$(\cos\theta, \sin\theta)$

$(-1,0)$   $(0,0)$

Figure 3: CWidth($\mathcal{A}$) for the set $\mathcal{A} := \{\mathcal{A}_\theta \cup -\mathcal{A}_\theta\}$ where $\mathcal{A}_\theta := \left\{ \binom{0}{0}, \binom{-1}{0}, \binom{\cos\theta}{\sin\theta} \right\}$ with $\theta \in (0, \pi/2]$

Figure 4: Ratio of theoretical and empirical rate for $\mathcal{A} := \{\mathcal{A}_\theta \cup -\mathcal{A}_\theta\}$ where $\mathcal{A}_\theta := \left\{ \binom{0}{0}, \binom{-1}{0}, \binom{\cos\theta}{\sin\theta} \right\}$ with $\theta \in (0, \pi/2)$ and 20 random target points $\binom{-\alpha_1}{\alpha_2}$ with $\alpha_i > 0$.

## A  Additional experiments

### A.1  An illustrative experiment: Tightness of Theorem 3

We now consider the setting depicted in Figure 3. We consider the set $\mathcal{A} := \{\mathcal{A}_\theta \cup -\mathcal{A}_\theta\}$ where $\mathcal{A}_\theta := \left\{ \binom{0}{0}, \binom{-1}{0}, \binom{\cos\theta}{\sin\theta} \right\}$ with $\theta \in (0, \pi/2)$. For this set CWidth($\mathcal{A}$) can be computed in closed form as CWidth($\mathcal{A}$) $= \sin(\theta/2)$. We then perform 20 runs of Algorithm 3 and report the ratio between the theoretical rate and the empirical one. The result is depicted in Figure 4. There, we considered an iteration starting from the origin minimizing the distance function to 20 random points $\binom{-\alpha_1}{\alpha_2}$ with $\alpha_i > 0$. The vertical bars shows minimal and maximal values.

### A.2  Real Data

**Hyperspectral image unmixing.** One of the classical applications of non-negative least squares are unmixing problems [11] such as hyperspectral image unmixing. Scalable unmixing approaches such as SPA [2] first extract a self-dictionary from a target image. Each pixel is then projected on the

Figure 5: Hyperspectral Imaging. We report suboptimality in a non-negative least squares task on real data, respectively.

Figure 6: SPA [2], VCA [36], XRAY [26], H2NMF [15], SNPA [13]

Figure 7: CBCL KL-Divergence

conic hull of the dictionary to estimate the abundance of each material. A standard technique is the hierarchical alternating least squares of [14] (nnlsHALSupdt). In Figure 5 (right), we compare the suboptimality of different methods as a function of the iteration. The dictionary is extracted from the undersampled Urban HSI Dataset[5] using SPA. This dataset contains 5,929 pixels, each associated with 162 hyperspectral features. The number of dictionary elements is 6, motivated by the fact that 6 different physical materials are depicted in this HSI data [16]. Therefore, FCMP converges after 6 iterations. For PWMP only $1.5\%$ of the iterations were bad steps on average for all dictionaries. Therefore, our corrective methods are proven to be competitive also on real data and the effect of the bad steps is negligible. We test other dictionaries for the Hyperspectral Imaging task. The result is depicted in Figure 6.

**KL-divergence non-negative low-rank matrix factorization.** The third task targets non-negative matrix factorization by minimization of the (non-least squares) KL-divergence-based objective function in Equation (3) in [20]. We again use the CBCL face dataset and we compare FCMP (Variant 1) and PWMP against the multiplicative algorithm from [31] (multiplicative) and the cyclic coordinate descent (CCD) from [20]. We use the same approximate LMO and parametrization of $\mathcal{A}$ as for the

| algorithm | training accuracy | test accuracy |
|---|---|---|
| NNMP | $0.8345 \pm 0.0242$ | $\mathbf{0.7419} \pm 0.0389$ |
| PWMP | $\mathbf{0.8379} \pm 0.0240$ | $\mathbf{0.7419} \pm 0.0392$ |
| FCMP | $0.8345 \pm 0.0238$ | $\mathbf{0.7419} \pm 0.0403$ |
| NNG | $0.8069 \pm 0.0518$ | $0.7258 \pm 0.0602$ |

Table 4: Logistic Regression with non-negative Garrote, median $\pm$ standard deviation. Our methods achieve highest accuracy.

| algorithm | relative error |
|---|---|
| multiplicative | 0.2991 |
| hals | 0.2927 |
| anls-as | 0.2912 |
| anls-bpp | 0.2914 |
| PWMP | 0.2913 |
| FCMP | **0.2909** |

Table 5: Non-negative tensor factorization on the KNIX dataset with rank 20

least-squares non-negative matrix factorization, and we set the $L$ to 0.1. The atom correction was implemented using the CCD algorithm. In this experiment the use of Variant 0 of FCMP is crucial as it allows for a much easier update step. The objective value as a function of the rank is depicted in Figure 7. We note that all the algorithms yield comparable objective value up to rank 35. For higher rank, FCMP and PWMP achieve a slightly smaller objective value.

**Least Squares Non-negative Tensor Factorization.** For this task we again use the KNIX dataset but now we arrange the scans to form a tensor of dimensionality $512 \times 512 \times 24$. We compare against the alternating nonnegativity-constrained least squares with block principal pivoting [24] (anlss-bpp) (which is also used in the FCMP and PWMP for the corrections), the active set method in [22] (anls-as), the hierarchical alternating least squares of [10] (hals) and the multiplicative updating algorithm (multiplicative) of [51]. The LMO for the tensor factorization is implemented with the tensor power method [1]. The result is depicted in Table 5.

In this section we showed that the merit of our algorithms is not limited to their theoretical properties. Indeed, our algorithms are competitive with several approaches and can be successfully used in a manifold of different tasks and datasets while not being tailored to any specific cost function.

## B  Affine Invariant Algorithms and Rates

In this section, present affine invariant versions of all presented algorithms, along with sub-linear and linear convergence guarantees. An optimization method is called *affine invariant* if it is invariant under linear or affine transformations of the input problem: If one chooses any re-parameterization of the domain $\mathcal{Q}$ by a *surjective* linear or affine map $\mathbf{M} : \hat{\mathcal{Q}} \rightarrow \mathcal{Q}$, then the "old" and "new" optimization problems $\min_{\mathbf{x} \in \mathcal{Q}} f(\mathbf{x})$ and $\min_{\hat{\mathbf{x}} \in \hat{\mathcal{Q}}} \hat{f}(\hat{\mathbf{x}})$ for $\hat{f}(\hat{\mathbf{x}}) := f(\mathbf{M}\hat{\mathbf{x}})$ look the same to the algorithm. We still require the set $\mathcal{Q}$ to contain the origin. In the following, we assume that after the transformation the origin is still on the border of $\mathrm{conv}(\mathcal{Q})$. If the origin is contained in the relative interior of $\mathrm{conv}(\mathcal{Q})$ we recover the existing affine invariant rates of [32].

### B.1  Affine invariant non-negative MP

To define an affine invariant upper bound on the objective function $f$, we use a variation of the affine invariant definition of the *curvature constant* from [21], adapted for MP in [32]:

$$C_{f,\mathcal{A}}^{\mathsf{MP}} := \sup_{\substack{\mathbf{s} \in \mathcal{A}, \, \mathbf{x} \in \mathrm{conv}(\mathcal{A}) \\ \gamma \in [0,1] \\ \mathbf{y} = \mathbf{x} + \gamma \mathbf{s}}} \frac{2}{\gamma^2} D(\mathbf{y}, \mathbf{x}), \tag{12}$$

where for cleaner exposition, we have used the shorthand notation $D(\mathbf{y}, \mathbf{x})$ to denote the difference of $f(\mathbf{y})$ and its linear approximation at $\mathbf{x}$

$$D(\mathbf{y}, \mathbf{x}) := f(\mathbf{y}) - f(\mathbf{x}) - \langle \mathbf{y} - \mathbf{x}, \nabla f(\mathbf{x}) \rangle.$$

Bounded curvature $C_{f,\mathcal{A}}$ closely corresponds to smoothness of the objective $f$. More precisely, if $\nabla f$ is $L$-Lipschitz continuous on $\mathrm{conv}(\mathcal{A})$ with respect to some arbitrary chosen norm $\|.\|$, i.e. $\|\nabla f(\mathbf{x}) - \nabla f(\mathbf{y})\|_* \leq L\|\mathbf{x} - \mathbf{y}\|$, where $\|.\|_*$ is the dual norm of $\|.\|$, then

$$C_{f,\mathcal{A}} \leq L\,\mathrm{radius}_{\|.\|}(\mathcal{A})^2 \;, \tag{13}$$

where $\mathrm{radius}_{\|.\|}(.)$ denotes the $\|.\|$-radius, see Lemma 15 in [32]. The curvature constant $C_{f,\mathcal{A}}$ is affine invariant as it does not depend on any norm. It combines the complexity of the domain $\mathrm{conv}(\mathcal{A})$ and the curvature of the objective function $f$ into a single quantity. Throughout this section, we assume availability of a finite constant $\rho > 0$ upper-bounding the atomic norms $\|.\|_{\mathcal{A}}$ of the optimum $\mathbf{x}^\star$, as well as the iterate sequence $(\mathbf{x}_t)_{t=0}^T$ until the current iteration, as defined in (6). We now present the affine invariant version of the non-negative MP algorithm (Algorithm 2) in Algorithm 5. The algorithm uses the curvature constant $C_{f,\rho(\mathcal{A}-\mathcal{A})}^{\mathsf{MP}}$ over the re-scaled set $\rho\,\mathrm{conv}(\mathcal{A} \cup -\mathcal{A})$, rather than $\mathrm{conv}(\mathcal{A} \cup -\mathcal{A})$.

---

**Algorithm 5** Affine Invariant Non-Negative Matching Pursuit

Same as Algorithm 2 except:
5:     $\gamma := \dfrac{\langle -\nabla f(\mathbf{x}_t), \rho^2 \mathbf{z}_t \rangle}{C_{f,\rho(\mathcal{A}-\mathcal{A})}^{\mathsf{MP}}}$

---

A sub-linear convergence guarantee for Algorithm 5 is presented in the following theorem.

**Theorem 5.** *Let $\mathcal{A} \subset \mathcal{H}$ be a bounded set with $\mathbf{0} \in \mathcal{A}$, $\rho := \max\{\|\mathbf{x}^\star\|_{\mathcal{A}}, \|\mathbf{x}_0\|_{\mathcal{A}}, \ldots, \|\mathbf{x}_T\|_{\mathcal{A}}\} < \infty$. Assume $f$ has smoothness constant $C_{f,\rho(\mathcal{A}\cup-\mathcal{A})}^{\mathsf{MP}}$. Then, Algorithm 5 converges for $t \geq 0$ as*

$$f(\mathbf{x}_t) - f(\mathbf{x}^\star) \leq \frac{4\left(\frac{2}{\delta} C_{f,\rho(\mathcal{A}\cup-\mathcal{A})}^{\mathsf{MP}} + \varepsilon_0\right)}{\delta t + 4},$$

*where $\delta \in (0, 1]$ is the relative accuracy parameter of the employed approximate LMO (3).*

Exact knowledge of $C_{f,\rho(\mathcal{A}\cup-\mathcal{A})}^{\mathsf{MP}}$ is not required; the same theorem also holds if any upper bound on $C_{f,\rho(\mathcal{A}\cup-\mathcal{A})}^{\mathsf{MP}}$ is used in the algorithm and resulting rate instead.

## B.2 Affine invariant corrective MP

An affine invariant version of AMP and PWMP, Algorithm 3, is presented in Algorithm 6. Note that Variant 1 of the fully corrective non-negative MP in Algorithm 4 is already affine invariant as it does not rely on any norm. Note that sublinear convergence is guaranteed with the rate indicated by Theorem 5 since each step of the affine invariant FCMP yields at least as much improvement as the affine invariant NNMP, Algorithm 5.

Since the update step in Algorithm 5 and the resulting upper bound on the progress in objective, based on the curvature constant (13), we used in the proof of Theorem 5 are not enough to ensure linear convergence, we use a different notion of curvature based on [27].

$$C_{f,\mathcal{A}}^{\mathsf{A}} = \sup_{\substack{\mathbf{s}\in\mathcal{A},\mathbf{x}\in\mathrm{conv}(\mathcal{A}) \\ \mathbf{v}\in\mathcal{S} \\ \gamma\in[0,1] \\ \mathbf{y}=\mathbf{x}+\gamma(\mathbf{s}-\mathbf{v})}} \frac{2}{\gamma^2} D(\mathbf{y}, \mathbf{x}).$$

The following positive step size quantity relates the dual certificate value of the descent direction

---

**Algorithm 6** Affine invariant AMP and PWMP

same as Algorithm 3 except for:
5:     $\gamma := \dfrac{\langle -\nabla f(\mathbf{x}_t), \rho^2 \mathbf{d}_t \rangle}{C_{f,\rho(\mathcal{A}-\mathcal{A})}^{\mathsf{A}}}$

---

$\mathbf{x}^\star - \mathbf{x}$ with the MP selected atom,

$$\gamma(\mathbf{x}, \mathbf{x}^\star) := \frac{\langle -\nabla f(\mathbf{x}), \mathbf{x}^\star - \mathbf{x} \rangle}{\langle -\nabla f(\mathbf{x}), \mathbf{s}(\mathbf{x}) - \mathbf{v}(\mathbf{x}) \rangle} \;, \tag{14}$$

for $\mathbf{s}(\mathbf{x}) := \arg\min_{\mathbf{s}\in\mathcal{A}} \langle \nabla f(\mathbf{x}), \mathbf{s}\rangle$ and $\mathbf{v}(\mathbf{x}) := \min_{\mathcal{S}\in\mathcal{S}_\mathbf{x}} \arg\max_{\mathbf{s}\in\mathcal{S}} \langle \nabla f(\mathbf{x}), \mathbf{s}\rangle$ where $\mathcal{S} \in \mathcal{S}_\mathbf{x}$ is the active set. We now define the affine invariant surrogate of strong convexity.

$$\mu_{f,\rho\mathcal{A}}^A := \inf_{\mathbf{x}\in\mathrm{conv}(\rho\mathcal{A})} \inf_{\substack{\mathbf{x}^\star\in\mathrm{conv}(\rho\mathcal{A}) \\ \langle\nabla f(\mathbf{x}),\mathbf{x}^\star-\mathbf{x}\rangle<0}} \frac{2}{\gamma(\mathbf{x},\mathbf{x}^\star)} D(\mathbf{x}^\star,\mathbf{x}).$$

**Theorem 6.** *Let $\mathcal{A} \subset \mathcal{H}$ be a bounded set containing the origin and let the objective function $f\colon \mathcal{H}\to\mathbb{R}$ have smoothness constant $C_{f,\rho(\mathcal{A}\cup-\mathcal{A})}^{\prime A}$ and strong convexity constant $\mu_{f,\rho\mathcal{A}}^A$*

*Then, the suboptimality of the iterates of Algorithm 3 and 4 decreases geometrically at each step in which $\gamma < \alpha_{\mathbf{v}_t}$ (henceforth referred to as "good steps") as:*

$$\varepsilon_{t+1} \leq (1-\beta)\,\varepsilon_t, \tag{15}$$

*where $\beta := \delta^2 \frac{\mu_{f,\rho\mathcal{A}}^A}{C_{f,\rho(\mathcal{A}\cup-\mathcal{A})}^A} \in (0,1]$, $\varepsilon_t := f(\mathbf{x}_t) - f(\mathbf{x}^\star)$ is the suboptimality at step $t$ and $\delta \in (0,1]$ is the relative accuracy parameter of the employed approximate LMO (Equation (3)). For AMP (Algorithm 3), $\beta^{\mathrm{AMP}} = \beta/2$. If $\mu_{f,\rho\mathcal{A}}^A = 0$ Algorithm 3 converges with rate $O(1/k(t))$ where $k(t)$ is the number of "good steps" up to iteration $t$.*

## C Proof of Lemma 1

If $\tilde{\mathbf{x}} \in \mathrm{cone}(\mathcal{A})$ and $\nabla f(\tilde{\mathbf{x}}) \notin T_\mathcal{A}$ then $\tilde{\mathbf{x}}$ is a solution to $\min_{\mathbf{x}\in\mathrm{cone}(\mathcal{A})} f(\mathbf{x})$.

*Proof.* We will prove this lemma by contradiction assuming that $\mathbf{x}^\star \neq \tilde{\mathbf{x}}$ and $\nabla f(\tilde{\mathbf{x}}) \notin T_\mathcal{A}$. Now, by convexity of $f$ we have that:

$$f(\mathbf{x}^\star) \geq f(\tilde{\mathbf{x}}) + \langle \nabla f(\tilde{\mathbf{x}}), \mathbf{x}^\star - \tilde{\mathbf{x}}\rangle$$

Since $\mathbf{x}^\star \neq \tilde{\mathbf{x}}$ we have also that $f(\mathbf{x}^\star) < f(\tilde{\mathbf{x}})$. Therefore:

$$0 < f(\tilde{\mathbf{x}}) - f(\mathbf{x}^\star) \leq \langle -\nabla f(\tilde{\mathbf{x}}), \mathbf{x}^\star - \tilde{\mathbf{x}}\rangle$$

which we rewrite as $\langle \nabla f(\tilde{\mathbf{x}}), \mathbf{x}^\star\rangle + \langle \nabla f(\tilde{\mathbf{x}}), -\tilde{\mathbf{x}}\rangle < 0$. Now we note that by the assumption that $\nabla f(\tilde{\mathbf{x}}) \notin T_\mathcal{A}$ we have that both these inner products are non negative which is absurd. To draw this conclusion note that $\mathbf{x}^\star \in \mathrm{cone}(\mathcal{A})$ we have that $\mathbf{x}^\star = \sum_i \alpha_i \mathbf{z}_i$ where $\mathbf{z}_i \in \mathcal{A}$ and $\alpha_i \geq 0\,\forall\, i$. $\qquad\square$

## D Sublinear Rates

**Theorem' 2.** *Let $\mathcal{A} \subset \mathcal{H}$ be a bounded set with $\mathbf{0} \in \mathcal{A}$, $\rho := \max\{\|\mathbf{x}^\star\|_\mathcal{A}, \|\mathbf{x}_0\|_\mathcal{A}, \ldots, \|\mathbf{x}_T\|_\mathcal{A},\}$ and $f$ be $L$-smooth over $\rho\,\mathrm{conv}(\mathcal{A}\cup-\mathcal{A})$. Then, Algorithms 2 and 4 with $\mathbf{x}_0 = \mathbf{0}$ converges for $t \geq 0$ as*

$$f(\mathbf{x}_t) - f(\mathbf{x}^\star) \leq \frac{4\left(\frac{2}{\delta}L\rho^2\,\mathrm{radius}(\mathcal{A})^2 + \varepsilon_0\right)}{\delta t + 4},$$

*where $\delta \in (0,1]$ is the relative accuracy parameter of the employed approximate LMO (3).*

*Proof.* We separately prove the convergence for the two algorithms.

**non-negative MP:** Recall that $\tilde{\mathbf{z}}_t$ is the atom returned by the inexact LMO after the comparison with $-\frac{\mathbf{x}_t}{\|\mathbf{x}_t\|_\mathcal{A}}$ at the current iteration $t$. We distinguish the two cases in which $\tilde{\mathbf{z}}_t \neq -\frac{\mathbf{x}_t}{\|\mathbf{x}_t\|_\mathcal{A}}$ (**case A**) and $\tilde{\mathbf{z}}_t = -\frac{\mathbf{x}_t}{\|\mathbf{x}_t\|_\mathcal{A}}$ (**case B**). Let us call $\bar{\mathcal{A}} := \mathcal{A} \cup \left\{-\frac{\mathbf{x}_t}{\|\mathbf{x}_t\|_\mathcal{A}}\right\}$. Note that $\mathrm{radius}(\bar{\mathcal{A}}) = \mathrm{radius}(\mathcal{A})$.

Recall that in the Algorithm the step size $\gamma$ is computed at each iteration via line search minimizing the quadratic upper bound on $f$ and no further clipping is made. The reason being that $f$ is convex, therefore, for $t > 0$ we have $f(\mathbf{x}_t) \leq f(\mathbf{0})$. Hence the minimum of $f$ over the line between $\mathbf{x}_t$ and the origin must lie between these two points making clipping unnecessary.

We start by upper bounding $f$ on $\rho \operatorname{conv}(\bar{\mathcal{A}})$ using smoothness as follows:

$$
\begin{aligned}
f(\mathbf{x}_{t+1}) &\leq \min_{\gamma \in \mathbb{R}_{\geq 0}} g_{\mathbf{x}_t}(\mathbf{x}_t + \gamma \tilde{\mathbf{z}}_t) \\
&= \min_{\gamma \in [0,1]} g_{\mathbf{x}_t}(\mathbf{x}_t + \gamma \rho \tilde{\mathbf{z}}_t) \\
&\leq \min_{\gamma \in [0,1]} f(\mathbf{x}_t) + \gamma \langle \nabla f(\mathbf{x}_t), \rho \tilde{\mathbf{z}}_t \rangle \\
&\quad + \frac{L}{2} \gamma^2 \rho^2 \|\tilde{\mathbf{z}}_t\|^2 \\
&\leq \min_{\gamma \in [0,1]} f(\mathbf{x}_t) + \gamma \langle \nabla f(\mathbf{x}_t), \rho \tilde{\mathbf{z}}_t \rangle \\
&\quad + \frac{L}{2} \gamma^2 \rho^2 \operatorname{radius}(\mathcal{A})^2
\end{aligned}
\tag{16}
$$

We now treat separately the linear term for **case A** and **case B**.

**case A:** We start from the definition of inexact LMO (Equation (3)). We then have:

$$
\langle \nabla f(\mathbf{x}_t), \tilde{\mathbf{z}}_t \rangle \leq \delta \langle \nabla f(\mathbf{x}_t), \mathbf{z}_t \rangle
$$

where $\mathbf{z}_t$ is the true minimizer of the linear problem (LMO). In other words, it holds that $\langle \nabla f(\mathbf{x}_t), \mathbf{z}_t \rangle \leq \langle \nabla f(\mathbf{x}_t), \mathbf{z} \rangle \; \forall \, \mathbf{z} \in \operatorname{conv}(\bar{\mathcal{A}})$ due to the $\arg\min$ in line 4 of Algorithm 2. Therefore, since $\mathbf{x}^\star \in \rho \operatorname{conv}(\bar{\mathcal{A}})$ it holds that:

$$
\langle \nabla f(\mathbf{x}_t), -\rho \mathbf{z}_t \rangle \geq \langle \nabla f(\mathbf{x}_t), -\mathbf{x}^\star \rangle.
$$

Using the same argument, since $-\frac{\mathbf{x}_t}{\|\mathbf{x}_t\|_{\mathcal{A}}} \in \bar{\mathcal{A}}$ and $\rho \geq \|\mathbf{x}_t\|_{\mathcal{A}}$, we have that $-\mathbf{x}_t \in \rho \operatorname{conv}(\bar{\mathcal{A}})$. Therefore:

$$
\langle \nabla f(\mathbf{x}_t), -\rho \mathbf{z}_t \rangle \geq \langle \nabla f(\mathbf{x}_t), \mathbf{x}_t \rangle.
$$

We can now bound the linear term of in (16) as:

$$
\begin{aligned}
\langle \nabla f(\mathbf{x}_t), -2\frac{\rho}{\delta} \tilde{\mathbf{z}}_t \rangle &= \langle \nabla f(\mathbf{x}_t), -\frac{\rho}{\delta} \tilde{\mathbf{z}}_t \rangle + \langle \nabla f(\mathbf{x}_t), -\frac{\rho}{\delta} \tilde{\mathbf{z}}_t \rangle \\
&\geq \langle \nabla f(\mathbf{x}_t), -\rho \mathbf{z}_t \rangle + \langle \nabla f(\mathbf{x}_t), -\rho \mathbf{z}_t \rangle \\
&\geq \langle \nabla f(\mathbf{x}_t), \mathbf{x}_t - \mathbf{x}^\star \rangle \\
&\geq f(\mathbf{x}_t) - f(\mathbf{x}^\star) =: \varepsilon_t
\end{aligned}
$$

where in the inequalities we used the the inexact oracle definition (see Section 2), the fact that both $-\mathbf{x}_t$ and $\mathbf{x}^\star \in \rho \operatorname{conv}(\bar{\mathcal{A}})$ and convexity respectively.

**case B:** Using line 4 of Algorithm 2 along with the inexact oracle definition we obtain:

$$
\langle \nabla f(\mathbf{x}_t), -\frac{\mathbf{x}_t}{\|\mathbf{x}_t\|_{\mathcal{A}}} \rangle \leq \delta \min_{\mathbf{z} \in \mathcal{A}} \langle \nabla f(\mathbf{x}_t), \mathbf{z} \rangle.
$$

Therefore, since $\mathbf{x}^\star \in \rho \operatorname{conv}(\mathcal{A})$ we can write:

$$
\begin{aligned}
\langle \nabla f(\mathbf{x}_t), -\frac{\rho}{\delta} \frac{\mathbf{x}_t}{\|\mathbf{x}_t\|_{\mathcal{A}}} \rangle &\leq \min_{\mathbf{z} \in \mathcal{A}} \langle \nabla f(\mathbf{x}_t), \rho \mathbf{z} \rangle \\
&\leq \langle \nabla f(\mathbf{x}_t), \mathbf{x}^\star \rangle
\end{aligned}
$$

We also have $\langle \nabla f(\mathbf{x}_t), -\mathbf{x}_t \rangle \leq 0$ and $\frac{\rho}{\delta \|\mathbf{x}_t\|_{\mathcal{A}}} > 1$, which yields:

$$
\langle \nabla f(\mathbf{x}_t), -\mathbf{x}_t \rangle \geq \langle \nabla f(\mathbf{x}_t), -\frac{\rho}{\delta} \frac{\mathbf{x}_t}{\|\mathbf{x}_t\|_{\mathcal{A}}} \rangle
$$

Putting these inequalites together we obtain:

$$
\begin{aligned}
\langle \nabla f(\mathbf{x}_t), \frac{2}{\delta} \rho \frac{\mathbf{x}_t}{\|\mathbf{x}_t\|_{\mathcal{A}}} \rangle &\geq \langle \nabla f(\mathbf{x}_t), \mathbf{x}_t \rangle + \max_{\mathbf{z} \in \mathcal{A}} \langle \nabla f(\mathbf{x}_t), -\rho \mathbf{z} \rangle \\
&\geq \langle \nabla f(\mathbf{x}_t), \mathbf{x}_t \rangle - \langle \nabla f(\mathbf{x}_t), \mathbf{x}^\star \rangle \\
&\geq \varepsilon_t
\end{aligned}
$$

**combining A and B**   By combining case A and case B we obtain:

$$f(\mathbf{x}_{t+1}) \le f(\mathbf{x}_t) + \min_{\gamma \in [0,1]} \left\{ -\frac{\delta}{2} \gamma \varepsilon_t + \frac{\gamma^2}{2} L \rho^2 \operatorname{radius}(\mathcal{A})^2 \right\}$$

Now, subtracting $f(\mathbf{x}^\star)$ from both sides and setting $C := L\rho^2 \operatorname{radius}(\mathcal{A})^2$, we get

$$\begin{aligned} \varepsilon_{t+1} \quad &\le \varepsilon_t + \min_{\gamma \in [0,1]} \left\{ -\frac{\delta}{2} \gamma \varepsilon_t + \frac{\gamma^2}{2} C \right\} \\ &\le \varepsilon_t - \frac{2}{\delta' t + 2} \delta' \varepsilon_t + \frac{1}{2} \left( \frac{2}{\delta' t + 2} \right)^2 C, \end{aligned}$$

where we set $\delta' := \delta/2$ and used $\gamma = \frac{2}{\delta' t + 2} \in [0,1]$ to obtain the second inequality. Finally, we show by induction

$$\varepsilon_t \le \frac{4\left( \frac{2}{\delta} C + \varepsilon_0 \right)}{t + 4} = 2 \frac{\left( \frac{1}{\delta'} C + \varepsilon_0 \right)}{\delta' t + 2}$$

for $t \ge 0$.

When $t = 0$ we get $\varepsilon_0 \le \left( \frac{1}{\delta'} C + \varepsilon_0 \right)$. Therefore, the base case holds. We now prove the induction step assuming $\varepsilon_t \le \frac{2\left( \frac{1}{\delta'} C + \varepsilon_0 \right)}{\delta' t + 2}$ as :

$$\begin{aligned} \varepsilon_{t+1} &\le \left( 1 - \frac{2\delta'}{\delta' t + 2} \right) \varepsilon_t + \frac{1}{2} C \left( \frac{2}{\delta' t + 2} \right)^2 \\ &\le \left( 1 - \frac{2\delta'}{\delta' t + 2} \right) \frac{2\left( \frac{1}{\delta'} C + \varepsilon_0 \right)}{\delta' t + 2} \\ &\quad + \frac{1}{2} \left( \frac{2}{\delta' t + 2} \right)^2 C + \frac{2}{(\delta' t + 2)^2} \delta' \varepsilon_0 \\ &= \frac{2\left( \frac{1}{\delta'} C + \varepsilon_0 \right)}{\delta' t + 2} \left( 1 - \frac{2\delta'}{\delta' t + 2} + \frac{\delta'}{\delta' t + 2} \right) \\ &\le \frac{2\left( \frac{1}{\delta'} C + \varepsilon_0 \right)}{\delta'(t+1) + 2}. \end{aligned}$$

Remembering that we set $C = L\rho^2 \operatorname{radius}(\mathcal{A})^2$ concludes the proof.

**Fully Corrective non-negative MP:**   The proof is trivial considering that:

$$f(\mathbf{x}_{t+1}) = \min_{\mathbf{x} \in \operatorname{cone}(\mathcal{S} \cup \mathbf{s}(\mathcal{A}, \mathbf{r}))} f(\mathbf{x}) \tag{17}$$

$$\le \min_{\mathbf{x} \in \operatorname{cone}(\mathcal{S} \cup \mathbf{s}(\mathcal{A}, \mathbf{r}))} g_{\mathbf{x}_t}(\mathbf{x}) \tag{18}$$

$$\le \min_{\gamma \in \mathbb{R}_{\ge 0}} g_{\mathbf{x}_t}(\mathbf{x}_t + \gamma \tilde{\mathbf{z}}_t) \tag{19}$$

where $\tilde{\mathbf{z}}_t \in A \cup \left\{ \frac{-\mathbf{x}_t}{\|\mathbf{x}_t\|_{\mathcal{A}}} \right\}$ as the search space in Equation (18) strictly contain the one in Equation (19). Equation (19) is also the beginning of the proof of the sublinear rate for NNMP which then concludes the proof. $\qquad\square$

## E   Linear Rate

**Theorem' 3.** *Let $\mathcal{A} \subset \mathcal{H}$ be a bounded set containing the origin and let the objective function $f \colon \mathcal{H} \to \mathbb{R}$ be both $L$-smooth and $\mu$-strongly convex over $\rho \operatorname{conv}(\mathcal{A} \cup -\mathcal{A})$.*

*Then, the suboptimality of the iterates of Algorithm 3 decreases geometrically at each step in which $\gamma < \alpha_{\mathbf{v}_t}$ (henceforth referred to as "good steps") as:*

$$\varepsilon_{t+1} \le (1 - \beta)\, \varepsilon_t, \tag{20}$$

*where $\beta := \delta^2 \frac{\mu \operatorname{CWidth}(\mathcal{A})^2}{L \operatorname{diam}(\mathcal{A})^2} \in (0,1]$, $\varepsilon_t := f(\mathbf{x}_t) - f(\mathbf{x}^\star)$ is the suboptimality at step $t$ and $\delta \in (0,1]$ is the relative accuracy parameter of the employed approximate LMO (Equation (3)). For AMP (Algorithm 3), $\beta^{\mathrm{AMP}} = \beta/2$. If $\mu = 0$ Algorithm 3 converges with rate $O(1/k(t))$ where $k(t)$ is the number of "good steps" up to iteration $t$.*

*Proof.* Let us consider the case of PWMP.

Consider the atoms $\tilde{\mathbf{z}}_t \in \mathcal{A}$ and $\tilde{\mathbf{v}}_t \in \mathcal{S}$ selected by the LMO at iteration $t$. Due to the smoothness property of $f$ it holds that:

$$f(\mathbf{x}_{t+1}) \leq \min_{\gamma \in \mathbb{R}} f(\mathbf{x}_t) + \gamma \langle \nabla f(\mathbf{x}_t), \tilde{\mathbf{z}}_t - \tilde{\mathbf{v}}_t \rangle$$
$$+ \frac{L}{2} \gamma^2 \|\tilde{\mathbf{z}}_t - \tilde{\mathbf{v}}_t\|^2.$$

for a good step (i.e. $\gamma < \alpha_{\mathbf{v}_t}$). Note that this also holds for variant 0 of Algorithm 4.

We minimize the upper bound with respect to $\gamma$ setting $\gamma = -\frac{1}{L} \langle \nabla f(\mathbf{x}_t), \frac{\tilde{\mathbf{z}}_t - \tilde{\mathbf{v}}_t}{\|\tilde{\mathbf{z}}_t - \tilde{\mathbf{v}}_t\|^2} \rangle$. Subtracting $f(\mathbf{x}^\star)$ from both sides and replacing the optimal $\gamma$ yields:

$$\varepsilon_{t+1} \leq \varepsilon_t - \frac{1}{2L} \left\langle \nabla f(\mathbf{x}_t), \frac{\tilde{\mathbf{z}}_t - \tilde{\mathbf{v}}_t}{\|\tilde{\mathbf{z}}_t - \tilde{\mathbf{v}}_t\|} \right\rangle^2 \tag{21}$$

Now writing the definition of strong convexity, we have the following inequality holding for all $\gamma \in \mathbb{R}$:

$$f(\mathbf{x}_t + \gamma(\mathbf{x}^\star - \mathbf{x}_t)) \geq f(\mathbf{x}_t) + \gamma \langle \nabla f(\mathbf{x}_t), \mathbf{x}^\star - \mathbf{x}_t \rangle +$$
$$\gamma^2 \frac{\mu}{2} \|\mathbf{x}^\star - \mathbf{x}_t\|^2$$

We now fix $\gamma = 1$ in the LHS and minimize with respect to $\gamma$ in the RHS:

$$\varepsilon_t \leq \frac{1}{2\mu} \left\langle \nabla f(\mathbf{x}_t), \frac{\mathbf{x}^\star - \mathbf{x}_t}{\|\mathbf{x}^\star - \mathbf{x}_t\|} \right\rangle^2$$

Combining this with (21) yields:

$$\varepsilon_t - \varepsilon_{t+1} \geq \frac{\mu}{L} \frac{\left\langle \nabla f(\mathbf{x}_t), \frac{\tilde{\mathbf{z}}_t - \tilde{\mathbf{v}}_t}{\|\tilde{\mathbf{z}}_t - \tilde{\mathbf{v}}_t\|} \right\rangle^2}{\left\langle \nabla f(\mathbf{x}_t), \frac{\mathbf{x}^\star - \mathbf{x}_t}{\|\mathbf{x}^\star - \mathbf{x}_t\|} \right\rangle^2} \varepsilon_t \tag{22}$$

We now use Theorem 8 to conclude the proof. For Away-steps MP the proof is trivially extended since $2 \min_{\mathbf{z} \in \mathcal{A} \cup -\mathcal{S}} \langle \nabla f(\mathbf{x}_t), \mathbf{z} \rangle \leq \min_{\mathbf{z} \in \mathcal{A}, \mathbf{v} \in \mathcal{S}} \langle \nabla f(\mathbf{x}_t), \mathbf{z} - \mathbf{v} \rangle$. Therefore, we obtain the same smoothness upper bound of the PWMP. The rest of the proof proceed as for PWMP with the additional $\frac{1}{2}$ factor.

**Sublinear Convergence for $\mu = 0$**   If $\mu = 0$ we have for PWMP:

$$f(\mathbf{x}_{t+1}) \leq \min_{\gamma \leq \alpha_{\mathbf{v}_t}} f(\mathbf{x}_t) + \gamma \langle \nabla f(\mathbf{x}_t), \tilde{\mathbf{z}}_t - \tilde{\mathbf{v}}_t \rangle \tag{23}$$
$$+ \frac{L}{2} \gamma^2 \|\tilde{\mathbf{z}}_t - \tilde{\mathbf{v}}_t\|^2. \tag{24}$$

which can be rewritten for a good step (i.e. no clipping is necessary) as:

$$\varepsilon_{t+1} \quad \leq \varepsilon_t + \min_{\gamma \in [0,1]} \left\{ -\frac{\delta}{2} \gamma \varepsilon_t + \frac{\gamma^2}{2} L \rho^2 \operatorname{diam}(\mathcal{A})^2 \right\}$$

using the same arguments of the proof of Theorem 2. Unfortunately, $\alpha_{\mathbf{v}_t}$ limits the improvement. On the other hand, we can repeat the induction of Theorem 2 for only the good steps. Therefore:

$$\varepsilon_{t+1} \quad \leq \varepsilon_t - \frac{2}{\delta' t + 2} \delta' \varepsilon_t + \frac{1}{2} \left( \frac{2}{\delta' t + 2} \right)^2 C,$$

where we set $\delta' := \delta/2$, $C = L \rho^2 \operatorname{diam}(\mathcal{A})^2$ and used $\gamma = \frac{2}{\delta' t + 2} \in [0,1]$ (since it is a good step this produce a valid upper bound). Finally, we show by induction

$$\varepsilon_t \leq \frac{4 \left( \frac{2}{\delta} C + \varepsilon_0 \right)}{t+4} = 2 \frac{\left( \frac{1}{\delta'} C + \varepsilon_0 \right)}{\delta' k(t) + 2}$$

where $k(t) \geq 0$ is the number of good steps at iteration $t$.

When $k(t) = 0$ we get $\varepsilon_0 \leq \left(\frac{1}{\delta'}C + \varepsilon_0\right)$. Therefore, the base case holds. We now prove the induction step assuming $\varepsilon_t \leq \frac{2\left(\frac{1}{\delta'}C + \varepsilon_0\right)}{\delta' k(t) + 2}$ as :

$$
\begin{aligned}
\varepsilon_{t+1} &\leq \left(1 - \frac{2\delta'}{\delta' k(t) + 2}\right) \varepsilon_t + \frac{1}{2} C \left(\frac{2}{\delta' k(t) + 2}\right)^2 \\
&\leq \left(1 - \frac{2\delta'}{\delta' k(t) + 2}\right) \frac{2\left(\frac{1}{\delta'}C + \varepsilon_0\right)}{\delta' k(t) + 2} \\
&\quad + \frac{1}{2} \left(\frac{2}{\delta' k(t) + 2}\right)^2 C + \frac{2}{(\delta' k(t) + 2)^2} \delta' \varepsilon_0 \\
&= \frac{2\left(\frac{1}{\delta'}C + \varepsilon_0\right)}{\delta' k(t) + 2} \left(1 - \frac{2\delta'}{\delta' k(t) + 2} + \frac{\delta'}{\delta' k(t) + 2}\right) \\
&\leq \frac{2\left(\frac{1}{\delta'}C + \varepsilon_0\right)}{\delta'(k(t) + 1) + 2}.
\end{aligned}
$$

For AFW the procedure is the same but the linear term of Equation 23 is divided by two. We proceed as before with the only difference that we call $\delta' = \delta/4$. $\qquad\square$

### E.1   Proof sketch for linear rate convergence of FCMP

**Theorem' 3.** *Let $\mathcal{A} \subset \mathcal{H}$ be a bounded set containing the origin and let the objective function $f \colon \mathcal{H} \to \mathbb{R}$ be both $L$-smooth and $\mu$-strongly convex over $\rho \operatorname{conv}(\mathcal{A} \cup -\mathcal{A})$.*

*Then, the suboptimality of the iterates of Algorithm 4 decreases geometrically at each step in which $\gamma < \alpha_{\mathbf{v}_t}$ (henceforth referred to as "good steps") as:*

$$
\varepsilon_{t+1} \leq (1 - \beta)\, \varepsilon_t, \tag{25}
$$

*where $\beta := \delta^2 \frac{\mu \operatorname{CWidth}(\mathcal{A})^2}{L \operatorname{diam}(\mathcal{A})^2} \in (0, 1]$, $\varepsilon_t := f(\mathbf{x}_t) - f(\mathbf{x}^\star)$ is the suboptimality at step $t$ and $\delta \in (0, 1]$ is the relative accuracy parameter of the employed approximate LMO (Equation (3)).*

*Proof.* The proof is trivial noticing that:

$$
\begin{aligned}
f(\mathbf{x}_{t+1}) &= \min_{\mathbf{x} \in \operatorname{cone}(\mathcal{S} \cup \mathbf{s}(\mathcal{A}, \mathbf{r}))} f(\mathbf{x}) \\
&\leq \min_{\mathbf{x} \in \operatorname{cone}(\mathcal{S} \cup \mathbf{s}(\mathcal{A}, \mathbf{r}))} g_{\mathbf{x}_t}(\mathbf{x}) \\
&\leq \min_{\gamma \leq \alpha_{\mathbf{v}_t}} g_{\mathbf{x}_t}(\mathbf{x}_t + \gamma(\mathbf{z}_t - \mathbf{v}_t))
\end{aligned}
$$

which is the beginning of the proof of Theorem 3. Note that there are no bad steps for variant 1. Since we minimize $f$ at each iteration, $\mathbf{v}_t$ is always zero and each step is unconstrained (i.e., no bad steps). $\qquad\square$

## F   Pyramidal Width

Let us first recall some definitions from [28].

**Directional Width**

$$
dirW(\mathcal{A}, \mathbf{r}) := \max_{\mathbf{s}, \mathbf{v} \in \mathcal{A}} \left\langle \frac{\mathbf{r}}{\|\mathbf{r}\|}, \mathbf{s} - \mathbf{v} \right\rangle \tag{26}
$$

**Pyramidal Directional Width**

$$
PdirW(\mathcal{A}, \mathbf{r}, \mathbf{x}) := \min_{\mathcal{S} \in \mathcal{S}_{\mathbf{x}}} dirW(\mathcal{S} \cup \{\mathbf{s}(\mathcal{A}, \mathbf{r})\}, \mathbf{r}) \tag{27}
$$

Where $\mathcal{S}_{\mathbf{x}} := \{\mathcal{S} \mid \mathcal{S} \subset \mathcal{A}$ such that $\mathbf{x}$ is a proper convex combination of all the elements in $\mathcal{S}\}$ and $\mathbf{s}(\mathcal{A}, \mathbf{r}) := \max_{\mathbf{s} \in \mathcal{A}} \langle \frac{\mathbf{r}}{\|\mathbf{r}\|}, \mathbf{s} \rangle$.

**Pyramidal Width**

$$PWidth(\mathcal{A}) := \min_{\substack{\mathcal{K}\in \text{faces}(\text{conv}(\mathcal{A})) \\ \mathbf{x}\in\mathcal{K} \\ \mathbf{r}\in\text{cone}(\mathcal{K}-\mathbf{x})\setminus\{\mathbf{0}\}}} PdirW(\mathcal{K}\cap\mathcal{A},\mathbf{r},\mathbf{x})$$

Inspired by the notion of pyramidal width we now define the cone width of a set $\mathcal{A}$.

**Cone Width**

$$\text{CWidth}(\mathcal{A}) := \min_{\substack{\mathcal{K}\in \text{g-faces}(\text{cone}(\mathcal{A})) \\ \mathbf{x}\in\mathcal{K} \\ \mathbf{r}\in\text{cone}(\mathcal{K}-\mathbf{x})\setminus\{\mathbf{0}\}}} PdirW(\mathcal{K}\cap\mathcal{A},\mathbf{r},\mathbf{x})$$

The linear rate analysis is dominated by the fact that, similarly as in FW, many step directions are constrained (the ones pointing outside of the cone). So these arguments are in line with [28] and the techniques are adapted here. Lemma 7 is a minor modification of [[28], Lemma 5], see also their Figure 3. If the gradient is not feasible, the vector with maximum inner product must lie on a facet. Furthermore, it has the same inner product with the gradient and with its orthogonal projection on that facet. While first proof of Lemma 7 follows [28], we also give a different proof which does not use the KKT conditions.

**Lemma 7.** *Let $\mathbf{x}$ be a reference point inside a polytope $\mathcal{K}\in \text{g-faces}(\text{cone}(\mathcal{A}))$ and $\mathbf{r}\in \text{lin}(\mathcal{K})$ is not a feasible direction from $\mathbf{x}$. Then, a feasible direction in $\mathcal{K}$ minimizing the angle with $\mathbf{r}$ lies on a facet $\mathcal{K}'$ of $\mathcal{K}$ that includes $\mathbf{x}$:*

$$\max_{\mathbf{e}\in\text{cone}(\mathcal{K}-\mathbf{x})}\langle\mathbf{r},\frac{\mathbf{e}}{\|\mathbf{e}\|}\rangle = \max_{\mathbf{e}\in\text{cone}(\mathcal{K}'-\mathbf{x})}\langle\mathbf{r},\frac{\mathbf{e}}{\|\mathbf{e}\|}\rangle$$

$$= \max_{\mathbf{e}\in\text{cone}(\mathcal{K}'-\mathbf{x})}\langle\mathbf{r}',\frac{\mathbf{e}}{\|\mathbf{e}\|}\rangle$$

*where $\mathbf{r}'$ is the orthogonal projection of $\mathbf{r}$ onto $\text{lin}(\mathcal{K}')$*

*Proof.* Let us center the problem in $\mathbf{x}$. We rewrite the optimization problem as:

$$\max_{\mathbf{e}\in\text{cone}(\mathcal{K}),\|\mathbf{e}\|=1}\langle\mathbf{r},\mathbf{e}\rangle$$

and suppose by contradiction that $\mathbf{e}$ is in the relative interior of the cone. By the KKT necessary conditions we have that $\mathbf{e}^\star$ is collinear with $\mathbf{r}$. Therefore $\mathbf{e}^\star = \pm\mathbf{r}$. Now we know that $\mathbf{r}$ is not feasible, therefore the solution is $\mathbf{e}^\star = -\mathbf{r}$. By Cauchy-Schwarz we know that this solution is minimizing the inner product which is absurd. Therefore, $\mathbf{e}^\star$ must lie on a face of the cone. The last equality is trivial considering that $\mathbf{r}'$ is the orthogonal projection of $\mathbf{r}$ onto $\text{lin}(\mathcal{K}')$.

**Alternative proof.** This proof extends the traditional proof technique of [28] to infinitely many constraints. We also reported the FW inspired proof for the readers that are more familiar with the FW analysis. Using proposition 2.11 of [8] (we also use their notation) the first order optimality condition minimizing a function $J$ in a general Hilbert space given a closed set $\mathcal{K}$ is that the directional derivative computed at the optimum $\bar{u}$ satisfy $J'(\bar{u})v \geq 0 \ \forall v \in \mathcal{T}(\mathcal{K}-\bar{u})$. Let us now assume that $\bar{u}$ is in the relative interior of $\mathcal{K}$. Then $\mathcal{T}(\mathcal{K}-\bar{u}) = \mathcal{H}$. Furthermore, $J'(\bar{u})v = \langle\mathbf{r},v\rangle$ which is clearly not greater or equal than zero for any element of $\mathcal{H}$. $\square$

Theorem 8 is the key argument to conclude the proof of Theorem 3 from Equation (22): we have to bound the ratio of those inner products with the cone width.

**Theorem 8.** *Let $\mathbf{r} = -\nabla f(\mathbf{x}_t)$, $\mathbf{x}\in\text{cone}(\mathcal{A})$, $\mathcal{S}$ be the active set and $\mathbf{z}$ and $\mathbf{v}$ obtained as in Algorithm 3. Then, using the notation from Lemma 7:*

$$\frac{\langle\mathbf{r},\mathbf{d}\rangle}{\langle\mathbf{r},\hat{\mathbf{e}}\rangle} \geq \text{CWidth}(\mathcal{A}) \tag{28}$$

*where $\mathbf{d} := \mathbf{z}-\mathbf{v}$, $\hat{\mathbf{e}} = \frac{\mathbf{e}}{\|\mathbf{e}\|}$ and $\mathbf{e} = \mathbf{x}^\star - \mathbf{x}_t$.*

*Proof.* As we already discussed we can consider $\mathbf{x} \in \text{conv}(\mathcal{A})$ instead of $\mathbf{x}_t \in \text{cone}(\mathcal{A})$ since both the cone and the set of feasible direction are invariant to a rescaling of $\mathbf{x}$ by a strictly positive constant. Let us center all the vectors in $\mathbf{x}$, then $\hat{\mathbf{e}}$ is just a vector with norm 1 in some face. As $\mathbf{x}$ is not optimal, by convexity we have that $\langle \mathbf{r}, \hat{\mathbf{e}} \rangle > 0$. By Cauchy-Schwartz we know that $\langle \mathbf{r}, \hat{\mathbf{e}} \rangle \leq \|\mathbf{r}\|$ since $\langle \mathbf{r}, \hat{\mathbf{e}} \rangle > 0$ and $\|\hat{\mathbf{e}}\| = 1$. By definition of $\mathbf{d}$ we have:

$$\langle \frac{\mathbf{r}}{\|\mathbf{r}\|}, \mathbf{d} \rangle = \max_{\mathbf{z} \in \mathcal{A}, \mathbf{v} \in \mathcal{S}} \langle \frac{\mathbf{r}}{\|\mathbf{r}\|}, \mathbf{z} - \mathbf{v} \rangle$$
$$\geq \min_{\mathcal{S} \subset \mathcal{S}_{\mathbf{x}}} \max_{\mathbf{z} \in \mathcal{A}, \mathbf{v} \in \mathcal{S}} \langle \frac{\mathbf{r}}{\|\mathbf{r}\|}, \mathbf{z} - \mathbf{v} \rangle$$
$$= PdirW(\mathcal{A}, \mathbf{r}, \mathbf{x}).$$

Now, if $\mathbf{r}$ is a feasible direction from $\mathbf{x}$ Equation (28) is proved (note that $PdirW(\mathcal{A}, \mathbf{r}, \mathbf{x}) \geq \text{CWidth}(\mathcal{A})$ as $\text{conv}(\mathcal{A}) \in \text{g-faces}(\text{cone}(\mathcal{A}))$ and $\text{conv}(\mathcal{A}) \cap \mathcal{A} = \mathcal{A}$). If $\mathbf{r}$ is not a feasible direction it means that $\mathbf{x}$ is on a face of $\text{cone}(\mathcal{A})$ and $\mathbf{r}$ points to the exterior of $\text{cone}(\mathcal{A})$ from $\mathbf{x}$. We then project $\mathbf{r}$ on the faces of $\text{cone}(\mathcal{A})$ containing $\mathbf{x}$ until it is a feasible direction. We start by lower bounding the ratio of the two inner products replacing $\hat{\mathbf{e}}$ with a vector of norm 1 in the cone that has maximum inner product with $\mathbf{r}$ (with abuse of notation we still call it $\hat{\mathbf{e}}$). We then write:

$$\frac{\langle \mathbf{r}, \mathbf{d} \rangle}{\langle \mathbf{d}, \hat{\mathbf{e}} \rangle} \geq \left( \max_{\mathbf{z} \in \mathcal{A}, \mathbf{v} \in \mathcal{S}} \langle \mathbf{r}, \mathbf{z} - \mathbf{v} \rangle \right) \cdot \left( \max_{\mathbf{e} \in \text{cone}(\mathcal{A} - \mathbf{x})} \langle \mathbf{r}, \frac{\mathbf{e}}{\|\mathbf{e}\|} \rangle \right)^{-1}$$

Let us assume that $\mathbf{r}$ is not feasible but without loss of generality is in $\text{lin}(\mathcal{A})$ since orthogonal components to $\text{lin}(\mathcal{A})$ does not influence the inner product with elements in $\text{lin}(\mathcal{A})$. Using Lemma 7 we know that:

$$\max_{\mathbf{e} \in \text{cone}(\mathcal{K} - \mathbf{x})} \langle \mathbf{r}, \frac{\mathbf{e}}{\|\mathbf{e}\|} \rangle = \max_{\mathbf{e} \in \text{cone}(\mathcal{K}' - \mathbf{x})} \langle \mathbf{r}, \frac{\mathbf{e}}{\|\mathbf{e}\|} \rangle$$
$$= \max_{\mathbf{e} \in \text{cone}(\mathcal{K}' - \mathbf{x})} \langle \mathbf{r}', \frac{\mathbf{e}}{\|\mathbf{e}\|} \rangle$$

Let us now consider the reduced cone $\text{cone}(\mathcal{K}')$ as $\mathbf{r} \in \text{lin}(\mathcal{K}')$. For the numerator we obtain:

$$\max_{\mathbf{z} \in \mathcal{A}, \mathbf{v} \in \mathcal{S}} \langle \mathbf{r}, \mathbf{z} - \mathbf{v} \rangle \overset{\mathcal{K}' \subset \mathcal{A}}{\geq} \max_{\mathbf{z} \in \mathcal{K}'} \langle \mathbf{r}, \mathbf{z} \rangle + \max_{\mathbf{v} \in \mathcal{S}} \langle -\mathbf{r}, \mathbf{v} \rangle$$

Putting numerator and denominator together we obtain:

$$\frac{\langle \mathbf{r}, \mathbf{d} \rangle}{\langle \mathbf{d}, \hat{\mathbf{e}} \rangle} \geq \left( \max_{\substack{\mathbf{z} \in \mathcal{K}' \\ \mathbf{v} \in \mathcal{S}}} \langle \mathbf{r}', \mathbf{z} - \mathbf{v} \rangle \right) \cdot \left( \max_{\mathbf{e} \in \text{cone}(\mathcal{K}' - \mathbf{x})} \langle \mathbf{r}', \frac{\mathbf{e}}{\|\mathbf{e}\|} \rangle \right)^{-1}$$

Note that $\mathcal{S} \subset \mathcal{K}'$. Indeed, $\mathbf{x}$ is a proper convex combination of the elements of $\mathcal{S}$ and $\mathbf{x} \in \mathcal{K}' \subset \text{conv}(\mathcal{A})$. Now if $\mathbf{r}'$ is a feasible direction in $\text{cone}(\mathcal{K}' - \mathbf{x})$ we obtain the cone width since $\text{cone}(\mathcal{K}')$ is a face of $\text{cone}(\mathcal{A})$. If not we reiterate the procedure projecting onto a lower dimensional face $\mathcal{K}''$. Eventually, we will obtain a feasible direction. Since $\langle \mathbf{r}, \hat{\mathbf{e}} \rangle \neq 0$ we will obtain $\mathbf{r}_{final} \neq \mathbf{0}$. $\quad\square$

**Lemma 4.** *If the origin is in the relative interior of $\text{conv}(\mathcal{A})$ with respect to its linear span, then $\text{cone}(\mathcal{A}) = \text{lin}(\mathcal{A})$ and $\text{CWidth}(\mathcal{A}) = \text{mDW}(\mathcal{A})$.*

*Proof.* Let us first rewrite the definition of cone width:

$$\text{CWidth}(\mathcal{A}) := \min_{\substack{\mathcal{K} \in \text{g-faces}(\text{cone}(\mathcal{A})) \\ \mathbf{x} \in \mathcal{K} \\ \mathbf{r} \in \text{cone}(\mathcal{K} - \mathbf{x}) \setminus \{\mathbf{0}\}}} PdirW(\mathcal{K} \cap \mathcal{A}, \mathbf{r}, \mathbf{x}).$$

The minimum is over all the feasible directions of the gradient from every point in the domain. It is not restrictive to consider $\mathbf{r}$ parallel to $\text{lin}(\mathcal{A})$ (because the orthogonal component has no influence). Therefore, from every point $\mathbf{x} \in \text{lin}(\mathcal{A})$ every $\mathbf{r} \in \text{lin}(\mathcal{A})$ is a feasible direction. The geometric constant then becomes:

$$\text{CWidth}(\mathcal{A}) = \min_{\substack{\mathcal{K} \in \text{g-faces}(\text{cone}(\mathcal{A})) \\ \mathbf{x} \in \mathcal{K} \\ \mathbf{r} \in \text{lin}(\mathcal{A}) \setminus \{\mathbf{0}\}}} PdirW(\mathcal{K} \cap \mathcal{A}, \mathbf{r}, \mathbf{x})$$

Let us now assume by contradiction that for any $\mathcal{K} \in$ g-faces we have:
$$\mathbf{0} \notin \underset{\mathbf{x}\in\mathcal{K}}{\arg\min} \, \underset{\mathbf{r}\in\text{lin}(\mathcal{A})\backslash\{\mathbf{0}\}}{\min} PdirW(\mathcal{K}\cap\mathcal{A},\mathbf{r},\mathbf{x}) \qquad (29)$$
Therefore, $\exists \mathbf{v} \in \mathcal{S}$ such that $\mathbf{v} \neq \mathbf{0}$ for any of the $\mathbf{x}$ minimizing (29). By definition, we have $\mathbf{0} \in \mathcal{S}$, which yields $\max_{\mathbf{v}\in\mathcal{S}}\langle \mathbf{r}, -\mathbf{v}\rangle \geq 0$ for every $\mathbf{r}$. Therefore, $\langle \mathbf{r}, \mathbf{z}-\mathbf{v}\rangle \geq \langle \mathbf{r}, \mathbf{z}\rangle$ which is absurd because we assumed zero was in the set of minimizers of (29). So $\mathbf{0}$ minimize the cone directional width which yields $\mathcal{S}_{\mathbf{x}} = \{\mathbf{0}\}$ and $\mathbf{v} = \mathbf{0}$. In conclusion we have:
$$\text{CWidth}(\mathcal{A}) = \underset{\mathbf{d}\in\text{lin}(\mathcal{A})}{\min} \underset{\mathbf{z}\in\mathcal{A}}{\max}\langle \frac{\mathbf{d}}{\|\mathbf{d}\|},\mathbf{z}\rangle = \text{mDW}(\mathcal{A})$$
$\square$

## G  Affine Invariant Sublinear Rate

**Theorem' 5.** *Let $\mathcal{A} \subset \mathcal{H}$ be a bounded set with $\mathbf{0} \in \mathcal{A}$, $\rho := \max\{\|\mathbf{x}^\star\|_\mathcal{A}, \|\mathbf{x}_0\|_\mathcal{A},\ldots,\|\mathbf{x}_T\|_\mathcal{A}\} < \infty$. Assume $f$ has smoothness constant $C^{MP}_{f,\rho(\mathcal{A}-\mathcal{A})}$. Then, Algorithm 5 converges for $t \geq 0$ as*
$$f(\mathbf{x}_t) - f(\mathbf{x}^\star) \leq \frac{4\left(\frac{2}{\delta}C^{MP}_{f,\rho(\mathcal{A}-\mathcal{A})} + \varepsilon_0\right)}{\delta t + 4},$$
*where $\delta \in (0,1]$ is the relative accuracy parameter of the employed approximate* LMO (3).

*Proof.* Recall that $\tilde{\mathbf{z}}_t$ is the atom returned by the inexact LMO after the comparison with $-\frac{\mathbf{x}_t}{\|\mathbf{x}_t\|_\mathcal{A}}$ at the current iteration $t$.

We start by upper bounding $f$ on $\rho \, \text{conv}(\bar{\mathcal{A}})$ using smoothness as follows:
$$f(\mathbf{x}_{t+1}) \leq \underset{\gamma\in[0,1]}{\min} f(\mathbf{x}_t) + \gamma\langle\nabla f(\mathbf{x}_t), \rho\tilde{\mathbf{z}}_t\rangle + \frac{\gamma^2}{2}C^{MP}_{f,\rho(\mathcal{A}-\mathcal{A})}$$
We now proceed bounding the linear term as done in the proof of Theorem 2 for **case A** and **case B** obtaining:
$$f(\mathbf{x}_{t+1}) \leq f(\mathbf{x}_t) + \underset{\gamma\in[0,1]}{\min}\left\{-\frac{\delta}{2}\gamma\varepsilon_t + \frac{\gamma^2}{2}C^{MP}_{f,\rho(\mathcal{A}-\mathcal{A})}\right\}$$
Now, subtracting $f(\mathbf{x}^\star)$ from both sides we get
$$\varepsilon_{t+1} \quad \leq \varepsilon_t + \min_{\gamma\in[0,1]}\left\{-\frac{\delta}{2}\gamma\varepsilon_t + \frac{\gamma^2}{2}C^{MP}_{f,\rho(\mathcal{A}-\mathcal{A})}\right\}$$
$$\leq \varepsilon_t - \frac{2}{\delta't+2}\delta'\varepsilon_t + \frac{1}{2}\left(\frac{2}{\delta't+2}\right)^2 C^{MP}_{f,\rho(\mathcal{A}-\mathcal{A})},$$
where we set $\delta' := \delta/2$ and used $\gamma = \frac{2}{\delta't+2} \in [0,1]$ to obtain the second inequality. Finally, we show by induction
$$\varepsilon_t \leq \frac{4\left(\frac{2}{\delta}C^{MP}_{f,\rho(\mathcal{A}-\mathcal{A})} + \varepsilon_0\right)}{t+4} = 2\frac{\left(\frac{1}{\delta'}C^{MP}_{f,\rho(\mathcal{A}-\mathcal{A})} + \varepsilon_0\right)}{\delta't + 2}$$
for $t \geq 0$.

When $t = 0$ we get $\varepsilon_0 \leq \left(\frac{1}{\delta'}C^{MP}_{f,\rho(\mathcal{A}-\mathcal{A})} + \varepsilon_0\right)$. Therefore, the base case holds. We now prove the induction step assuming $\varepsilon_t \leq \frac{2\left(\frac{1}{\delta'}C^{MP}_{f,\rho(\mathcal{A}-\mathcal{A})}+\varepsilon_0\right)}{\delta't+2}$ as :
$$\varepsilon_{t+1} \leq \left(1 - \frac{2\delta'}{\delta't+2}\right)\varepsilon_t + \frac{1}{2}C^{MP}_{f,\rho(\mathcal{A}-\mathcal{A})}\left(\frac{2}{\delta't+2}\right)^2$$
$$\leq \left(1 - \frac{2\delta'}{\delta't+2}\right)\frac{2\left(\frac{1}{\delta'}C^{MP}_{f,\rho(\mathcal{A}-\mathcal{A})}+\varepsilon_0\right)}{\delta't+2}$$
$$+ \frac{1}{2}\left(\frac{2}{\delta't+2}\right)^2 C^{MP}_{f,\rho(\mathcal{A}-\mathcal{A})} + \frac{2}{(\delta't+2)^2}\delta'\varepsilon_0$$
$$= \frac{2\left(\frac{1}{\delta'}C^{MP}_{f,\rho(\mathcal{A}-\mathcal{A})}+\varepsilon_0\right)}{\delta't+2}\left(1 - \frac{2\delta'}{\delta't+2} + \frac{\delta'}{\delta't+2}\right)$$
$$\leq \frac{2\left(\frac{1}{\delta'}C^{MP}_{f,\rho(\mathcal{A}-\mathcal{A})}+\varepsilon_0\right)}{\delta'(t+1)+2}.$$

$\square$

We next explore the relationship of $C^{\mathrm{MP}}_{f,\rho(\mathcal{A}\cup-\mathcal{A})}$ and the smoothness parameter. Recall that $f$ is *L-smooth* with respect to a given norm $\|.\|$ over a set $\mathcal{Q}$ if

$$\|\nabla f(\mathbf{x}) - \nabla f(\mathbf{y})\|_* \le L\|\mathbf{x}-\mathbf{y}\| \text{ for all } \mathbf{x},\mathbf{y} \in \mathcal{Q}, \tag{30}$$

where $\|.\|_*$ is the dual norm of $\|.\|$.

**Lemma 9.** *Assume $f$ is L-smooth with respect to a given norm $\|.\|$, over the set $\mathrm{conv}(\mathcal{A})$. Then,*

$$C^{\mathrm{MP}}_{f,\rho(\mathcal{A}\cup-\mathcal{A})} \le L\,\rho^2\,\mathrm{radius}_{\|.\|}(\mathcal{A})^2 \tag{31}$$

*Proof.* By the definition of smoothness of $f$ with respect to $\|.\|$,

$$D(\mathbf{y},\mathbf{x}) \le \frac{L}{2}\|\mathbf{y}-\mathbf{x}\|^2.$$

Hence, from the definition of $C^{\mathrm{MP}}_{f,\rho(\mathcal{A}\cup-\mathcal{A})}$,

$$
\begin{aligned}
C^{\mathrm{MP}}_{f,\mathcal{A}} &\le \sup_{\substack{\mathbf{s}\in\rho\mathcal{A},\mathbf{x}\in\mathrm{conv}(\rho\mathcal{A})\\ \gamma\in[0,1]\\ \mathbf{y}=\mathbf{x}+\gamma\mathbf{s}}} \frac{2}{\gamma^2}\frac{L}{2}\|\mathbf{y}-\mathbf{x}\|^2 \\
&= L\rho^2\sup_{\mathbf{s}\in\mathcal{A}}\|\mathbf{s}\|^2 \\
&= L\,\rho^2\,\mathrm{radius}_{\|.\|}(\mathcal{A})^2 .
\end{aligned}
$$

$\square$

## H  Affine Invariant Linear Rate

**Theorem' 6.** *Let $\mathcal{A}\subset\mathcal{H}$ be a bounded set containing the origin and let the objective function $f\colon\mathcal{H}\to\mathbb{R}$ have smoothness constant $C^{\mathrm{A}}_{f,\rho(\mathcal{A}\cup-\mathcal{A})}$ and strong convexity constant $\mu^{A}_{f,\rho\mathcal{A}}$*

*Then, the suboptimality of the iterates of Algorithm 3 and 4 decreases geometrically at each step in which $\gamma < \alpha_{\mathbf{v}_t}$ (henceforth referred to as "good steps") as:*

$$\varepsilon_{t+1} \le (1-\beta)\,\varepsilon_t, \tag{32}$$

*where $\beta := \delta^2 \frac{\mu^{A}_{f,\rho\mathcal{A}}}{C^{\mathrm{A}}_{f,\rho(\mathcal{A}\cup-\mathcal{A})}} \in (0,1]$, $\varepsilon_t := f(\mathbf{x}_t) - f(\mathbf{x}^\star)$ is the suboptimality at step $t$ and $\delta\in(0,1]$ is the relative accuracy parameter of the employed approximate $\mathrm{LMO}$ (Equation (3)). For AMP (Algorithm 3), $\beta^{\mathrm{AMP}} = \beta/2$. If $\mu^{A}_{f,\rho\mathcal{A}} = 0$ Algorithm 3 converges with rate $O(1/k(t))$ where $k(t)$ is the number of "good steps" up to iteration $t$.*

*Proof.* Let us first consider the PWMP update. Using the definition of $C^{\mathrm{A}}_{f,\rho(\mathcal{A}\cup-\mathcal{A})}$ we upper-bound $f$ on $\rho\,\mathrm{conv}(\mathcal{A})$ as follows

$$
\begin{aligned}
f(\mathbf{x}_{t+1}) &\le \min_{\gamma\in[0,1]} f(\mathbf{x}_t) + \gamma\langle\nabla f(\mathbf{x}_t),\rho\tilde{\mathbf{z}}_t - \rho\tilde{\mathbf{v}}_t\rangle \\
&\quad + \frac{\gamma^2}{2}C^{\mathrm{A}}_{f,\rho(\mathcal{A}\cup-\mathcal{A})} \\
&= \min_{\gamma\in\mathbb{R}} f(\mathbf{x}_t) + \gamma\langle\nabla f(\mathbf{x}_t),\rho\tilde{\mathbf{z}}_t - \rho\tilde{\mathbf{v}}_t\rangle \\
&\quad + \frac{\gamma^2}{2}C^{\mathrm{A}}_{f,\rho(\mathcal{A}\cup-\mathcal{A})} \\
&= f(\mathbf{x}_t) - \frac{\rho^2}{2C^{\mathrm{A}}_{f,\rho(\mathcal{A}\cup-\mathcal{A})}}\langle\nabla f(\mathbf{x}_t),\tilde{\mathbf{z}}_t - \tilde{\mathbf{v}}_t\rangle^2 .
\end{aligned}
$$

This upper bound holds for Algorithm 6 every time $\rho\gamma < \alpha_{\mathbf{v}}$ as $\rho\gamma$ minimizing the RHS of the first equality coincides with the update of Algorithm 5 Line 5. The first equality holds as $C^{\mathrm{A}}_{f,\rho(\mathcal{A}\cup-\mathcal{A})}$

is defined on $\rho \operatorname{conv}(\mathcal{A})$ and $\rho \operatorname{conv}(\mathcal{A})$ contains all iterates by definition, so that the unconstrained minimum lies in $[0, 1]$ assuming $\rho\gamma < \alpha_{\mathbf{v}}$.

Using $\varepsilon_t = f(\mathbf{x}^\star) - f(\mathbf{x}_t)$, we can lower bound the error decay as

$$\varepsilon_t - \varepsilon_{t+1} \geq \frac{\rho^2}{2C^{\mathrm{A}}_{f,\rho(\mathcal{A}\cup -\mathcal{A})}} \left\langle \nabla f(\mathbf{x}_t), \tilde{\mathbf{z}}_t - \tilde{\mathbf{v}}_t \right\rangle^2. \tag{33}$$

Starting from the definition of $\mu^A_{f,\rho\mathcal{A}}$ we get,

$$
\begin{aligned}
\frac{\gamma(\mathbf{x}_t, \mathbf{x}^\star)^2}{2} \mu^A_{f,\rho\mathcal{A}} &\leq\ f(\mathbf{x}^\star) - f(\mathbf{x}_t) - \langle \nabla f(\mathbf{x}_t), \mathbf{x}^\star - \mathbf{x}_t \rangle \\
&=\ -\varepsilon_t \\
&\ +\ \gamma(\mathbf{x}_t, \mathbf{x}^\star)\langle -\nabla f(\mathbf{x}_t), \mathbf{s}(\mathbf{x}_t) - \mathbf{v}(\mathbf{x}) \rangle,
\end{aligned}
$$

which gives

$$
\begin{aligned}
\varepsilon_t &\leq\ -\frac{\gamma(\mathbf{x}_t, \mathbf{x}^\star)^2}{2} \mu^A_{f,\rho\mathcal{A}} \\
&\ +\ \gamma(\mathbf{x}_t, \mathbf{x}^\star)\langle -\nabla f(\mathbf{x}_t), \mathbf{s}(\mathbf{x}_t) - \mathbf{v}(\mathbf{x}) \rangle \\
&\leq\ \frac{\langle -\nabla f(\mathbf{x}_t), \mathbf{s}(\mathbf{x}_t) - \mathbf{v}(\mathbf{x}) \rangle^2}{2\mu^A_{f,\rho\mathcal{A}}} \\
&=\ \frac{\langle -\nabla f(\mathbf{x}_t), \rho(\tilde{\mathbf{z}}_t - \tilde{\mathbf{v}}_t) \rangle^2}{2\delta^2 \mu^A_{f,\rho\mathcal{A}}}
\end{aligned}
\tag{34}
\tag{35}
$$

where the last inequality is by the quality of the approximate LMO as used in the algorithm, as defined in (3).

Combining equations (33) and (35), we have

$$\varepsilon_t - \varepsilon_{t+1} \geq \delta^2 \frac{\mu^A_{f,\rho\mathcal{A}}}{C^{\mathrm{A}}_{f,\rho(\mathcal{A}\cup -\mathcal{A})}}\, \varepsilon_t,$$

which proves the claimed result. The proof for AMP and FCMP follows directly using the same argument used in the proof of Theorem 3. The upper bound used in the FCMP is the affine invariant notion of smoothness. The proof steps for the sublinear convergence is the same as the one of Theorem 3 replacing $C$ with $C^{\mathrm{A}}_{f,\rho(\mathcal{A}\cup -\mathcal{A})}$. $\qquad\square$

**Lemma 10.** *If $f$ is $\mu$ strongly convex over the domain $\operatorname{conv}(\rho\mathcal{A})$ with respect to some arbitrary cholsen norm $\|\cdot\|$, then*

$$\mu^A_{f,\rho\mathcal{A}} \geq \mu\,\mathrm{CWidth}(\mathcal{A})^2$$

*Proof.* From the strong convexity:

$$
\begin{aligned}
\mu^A_{f,\rho\mathcal{A}} &=\ \inf_{\mathbf{x}\in\operatorname{conv}(\rho\mathcal{A})}\ \inf_{\substack{\mathbf{x}^\star\in\operatorname{conv}(\rho\mathcal{A}) \\ \langle \nabla f(\mathbf{x}), \mathbf{x}^\star - \mathbf{x}\rangle < 0}} \frac{2}{\gamma(\mathbf{x},\mathbf{x}^\star)} D(\mathbf{x}^\star, \mathbf{x}) \\
&\geq\ \inf_{\substack{\mathbf{x},\mathbf{x}^\star\in\operatorname{conv}(\rho\mathcal{A}), \\ \langle -\nabla f(\mathbf{x}), \mathbf{x}^\star - \mathbf{x}\rangle > 0}} \mu \left( \frac{\langle -\nabla f(\mathbf{x}), \mathbf{s}(\mathbf{x}) - \mathbf{v}(\mathbf{x}) \rangle}{\langle -\nabla f(\mathbf{x}), \frac{\mathbf{x}^\star - \mathbf{x}}{\|\mathbf{x}^\star - \mathbf{x}\|_\mathcal{A}}\rangle} \right)^2 \\
&\geq\ \mu\,\mathrm{CWidth}(\mathcal{A})^2
\end{aligned}
$$

where in the last inequality we used Theorem 8.

The proof for away-steps uses the same argument we used in the norm based rate. $\qquad\square$

**Lemma 11.** *Assume $f$ is $L$-smooth with respect to a given norm $\|.\|$, over the set $\operatorname{conv}(\mathcal{A})$. Then,*

$$C^{\mathrm{A}}_{f,\rho(\mathcal{A}\cup -\mathcal{A})} \leq L\,\rho^2\,\mathrm{diam}_{\|.\|}(\mathcal{A})^2 \tag{36}$$

*Proof.* By the definition of smoothness of $f$ with respect to $\|.\|$,

$$D(\mathbf{y}, \mathbf{x}) \leq \frac{L}{2} \|\mathbf{y} - \mathbf{x}\|^2.$$

Hence, from the definition of $C^{\mathsf{MP}}_{f,\rho(\mathcal{A} \cup -\mathcal{A})}$,

$$
\begin{aligned}
C^{\mathsf{MP}}_{f,\mathcal{A}} \quad &\leq \quad \sup_{\substack{\mathbf{s} \in \rho\mathcal{A}, \mathbf{x} \in \mathrm{conv}(\rho\mathcal{A}) \\ \mathbf{v} \in \mathcal{S} \\ \gamma \in [0,1] \\ \mathbf{y} = \mathbf{x} + \gamma(\mathbf{s} - \mathbf{v})}} \frac{2}{\gamma^2} \frac{L}{2} \|\mathbf{y} - \mathbf{x}\|^2 \\
&= \quad L\rho^2 \sup_{\substack{\mathbf{x} \in \mathrm{conv}(\rho\mathcal{A}) \\ \mathbf{s} \in \mathcal{A} \\ \mathbf{v} \in \mathcal{S}}} \|\mathbf{s} - \mathbf{v}\|^2 \\
&= \quad L\,\rho^2\,\mathrm{diam}_{\|.\|}(\mathcal{A})^2 \,.
\end{aligned}
$$

□