[Reviews · NeurIPS 2017]

Reviewer 1



%%%% Summary The authors consider constrained convex optimization problems where the constraints are conic and given by a bounded set of atoms. The authors propose linear oracle based method to tackle such problems in the spirit of Frank-Wolfe algorithm and Matching Pursuit techniques. The main algorithm is the Non-Negative Matching Pursuit and the authors propose several active set variants. The paper contains convergence analysis for all the algorithms under different scenarios. In a nutshel, the convergence rate is sublinear for general objectives and linear for strongly convex objectives. The linear rates involve a new geometric quantity, the cone width. Finally the authors illustrate the relevance of their algorithm on several machine learning tasks and different datasets. %%%% Main comments I did not have time to check the affine invariant algorithms and analyses presented in the appendix. The papers contains interesting novel ideas and extensions for linear oracle based optimization methods but the technical presentation suffer some weaknesses: - Theorem 2 which presentation is problematic and does not really provide any convergence guaranty. - All the linear convergence rates rely on Theorem 8 which is burried at the end of the appendix and which proof is not clear enough. - Lower bounds on the number of good steps of each algorithm which are not really proved since they rely on an argument of the type "it works the same as in another close setting". The numerical experiments are numerous and convincing, but I think that the authors should provide empirical evidences showing that the computational cost are of the same order of magnitude compared competing methods for the experiments they carried out. %%%% Details on the main comments %% Theorem 2 The presention and statement of Theorem 2 (and all the sublinear rates given in the paper) has the following form: - Given a fixed horizon T - Consider rho, a bound on the iterates x_0 ... x_T - Then for all t > 0 the suboptimality is of the order of c / t where c depends on rho. First, the proof cannot hold for all t > 0 but only for 0 < t <= T. Indeed, in the proof, equation (16) relies on the fact that the rho bound holds for x_t which is only ensured for t < = T. Second the numerator actually contains rho^2. When T increases, rho could increase as well and the given bound does not even need to approach 0. This presentation is problematic. One possible way to fix this would be to provide a priori conditions (such as coercivity) which ensure that the sequence of iterates remain in a compact set, allowing to define an upper bound independantly of the horizon T. In the proof I did not understand the sentence "The reason being that f is convex, therefore, for t > 0 we have f (x t ) < = f (0)." %% Lemma 7 and Theorem 8 I could not understand Lemma 7. The equation is given without any comment and I cannot understand its meaning without further explaination. Is this equation defining K'? Or is it the case that K' can be chosen to satisfy this equation? Does it have any other meaning? Lemma 7 deals only with g-faces which are polytopes. Is it always the case? What happens if K is not a polytope? Can this be done without loss of generality? Is it just a typo? Theorem 8: The presentation is problematic. In Lemma 7, r is not a feasible direction. In Theorem 8, it is the gradient of f at x_t. Theorem 8 says "using the notation from Lemma 7". The proof of Theorem 8 says "if r is a feasible direction". All this makes the work of the reader very hard. Notations of Lemma 7 are not properly used: - What is e? e is not fixed by Lemma 7, it is just a variable defining a maximum. This is a recurent mistake in the proofs. - What is K? K is supposed to be given in Lemma 7 but not in Theorem 8. - Polytope? All this could be more explicit. "As x is not optimal by convexity we have that < r , e > > 0". Where is it assumed that $x$ is not optimal? How does this translate in the proposed inequality? What does the following mean? "We then project r on the faces of cone(A) containing x until it is a feasible direction" Do the author project on an intersection of faces or alternatively on each face or something else? It would be more appropriate to say "the projection is a feasible direction" since r is fixed to be the gradient of f. It is very uncomfortable to have the value of r changing within the proof in an "algorithmic fashion" and makes it very hard to check accuracy of the arguments. In any case, I suspect that the resulting r could be 0 in which case the next equation does not make sense. What prevents the resulting r from being null? In the next sentences, the authors use Lemma 7 which assumes that r is not a feasible direction. This is contradictory with the preceeding paragraph. At this point I was completely confused and lost hope to understand the details of this proof. What is r' on line 723 and in the preceeding equation? I understand that there is a kind of recursive process in the proof. Why should the last sentence be true? %% Further comments Line 220, max should be argmax I did not really understand the non-negative matrix facotrization experiment. Since the resulting approximation is of rank 10, does it mean that the authors ran their algorithm for 10 steps only?

Reviewer 2



This paper considers the optimization problems over the convex cone, parameterized as the conic hull of a possibly infinite atom set. The authors established optimization convergence results for the proposed algorithms which apply to general atom sets and objective functions. Overall, the paper is well written and clearly states the contributions and the connections to existing literatures. 1. The paper targets an interesting topic, namely the parameterization of the optimization problems over the conic hull of a possibly infinite atom set, which is between but different from what matching pursuit over the linear span of a finite atom set and frank-wolfe over the convex hull of a set of atoms. This facilitates the proposal of the algorithms with optimization theoretical results for general objective functions and atom sets. 2. The paper fills in an empty space of showing the optimization convergence rates of greedy algorithms in matching pursuit and frank wolfe because the atom set is not guaranteed to contain an atom aligned with a descent direction for all possible suboptimal iterates. This shed light on understanding these problems. 3. The authors also introduced away-step, pairwise and fully corrective MP variants with theoretical guarantees for completeness.

Reviewer 3



In this paper, authors target an “intermediate case” between the two domain parameterizations given by the linear span and the convex hull of an atom set, that is the parameterization of the optimization domain as the conic hull of a possibly infinite atom set. This is different from the greedy optimization methods such as matching pursuit and Frank-Wolfe algorithms over the line span and the convex hull of a set of atoms, respectively. They give explicit convergence rates for non-negative MP algorithms. The alignment assumption of the existing MP method does not hold if the optimization domain is a cone. Authors present modifications of existing non-negative variants of MP so as to corroborate this issue along with the resulting MP-type algorithms for conic problems and corresponding convergence guarantees. Authors proposed three algorithms with different variants. It might be interesting to see which algorithm is the preferred in what conditions. It is useful to participants to apply the best method to their problems. In lines 202-205, authors pointed out that one has to take either the risk of choosing tau too small and failing to recover an optimal solution, or rely on too large tau, which can result in slow convergence. Does it mean that a large tau is preferred for the convergence guarantee without considering the time complexity? To verify the proposed algorithms, authors conducted experiments on three problems. However, the descriptions of the experimental settings and analysis of the results are too rough. It is better to have detailed discussion. Moreover, the computational time might be an interesting evaluation criterion for comparing different methods as well as their variants.